# The role of adolescent lifestyle habits in biological aging: A prospective twin study

**Anna Kankaanpää[1]\*, Asko Tolvanen[2], Aino Heikkinen[3], Jaakko Kaprio[3], Miina Ollikainen[3], Elina Sillanpää[1,3]**

[1]Gerontology Research Center (GEREC), Faculty of Sport and Health Sciences, University of Jyväskylä, Jyväskylä, Finland; [2]Methodology Center for Human Sciences, University of Jyväskylä, Jyväskylä, Finland; [3]Institute for Molecular Medicine Finland (FIMM), HiLife, University of Helsinki, Helsinki, Finland

## Abstract

**Background:** Adolescence is a stage of fast growth and development. Exposures during puberty can have long-term effects on health in later life. This study aims to investigate the role of adolescent lifestyle in biological aging.

**Methods:** The study participants originated from the longitudinal FinnTwin12 study (n = 5114). Adolescent lifestyle-related factors, including body mass index (BMI), leisure-time physical activity, smoking, and alcohol use, were based on self-reports and measured at ages 12, 14, and 17 years. For a subsample, blood-based DNA methylation (DNAm) was used to assess biological aging with six epigenetic aging measures in young adulthood (21–25 years, n = 824). A latent class analysis was conducted to identify patterns of lifestyle behaviors in adolescence, and differences between the subgroups in later biological aging were studied. Genetic and environmental influences on biological aging shared with lifestyle behavior patterns were estimated using quantitative genetic modeling.

**Results:** We identified five subgroups of participants with different adolescent lifestyle behavior patterns. When DNAm GrimAge, DunedinPoAm, and DunedinPACE estimators were used, the class with the unhealthiest lifestyle and the class of participants with high BMI were biologically older than the classes with healthier lifestyle habits. The differences in lifestyle-related factors were maintained into young adulthood. Most of the variation in biological aging shared with adolescent lifestyle was explained by common genetic factors.

**Conclusions:** These findings suggest that an unhealthy lifestyle during pubertal years is associated with accelerated biological aging in young adulthood. Genetic pleiotropy may largely explain the observed associations.

**Funding:** This work was supported by the Academy of Finland (213506, 265240, 263278, 312073 to J.K., 297908 to M.O. and 341750, 346509 to E.S.), EC FP5 GenomEUtwin (J.K.), National Institutes of Health/National Heart, Lung, and Blood Institute (grant HL104125), EC MC ITN Project EPITRAIN (J.K. and M.O.), the University of Helsinki Research Funds (M.O.), Sigrid Juselius Foundation (J.K. and M.O.), Yrjö Jahnsson Foundation (6868), Juho Vainio Foundation (E.S.) and Päivikki and Sakari Sohlberg foundation (E.S.).

**\*For correspondence:** anna.k.kankaanpaa@jyu.fi

**Competing interest:** The authors declare that no competing interests exist.

## Editor's evaluation

This is an important article that is methodologically compelling that provides evidence that an unhealthy lifestyle during adolescence accelerates epigenetic age in adulthood and that these associations are largely explained by the effect of shared genetic influences. The main strengths of this

article are the relatively large sample size, longitudinal assessment of lifestyle factors, and sophisticated statistical analyses. The article will be of interest to a broad audience, including individuals working on methylation, epidemiology, and/or aging.

## Introduction

Epidemiological studies of life course have indicated that exposures during early life have long-term effects on later health (*Kuh et al., 2003*). Unhealthy environments and lifestyle habits during rapid cell division can affect the structure or functions of organs, tissues, or body systems, and these changes can subsequently affect health and disease in later life (*Biro and Deardorff, 2013*; *Power et al., 2013*). For example, lower birth weight and fast growth during childhood predispose individuals to coronary heart disease and increased blood pressure in adulthood (*Osmond and Barker, 2000*). In addition to infancy and childhood, adolescence is also a critical period of growth.

Adolescence is characterized by pubertal maturation and growth spurts. Early pubertal development is linked to worse health conditions, such as obesity and cardiometabolic risk factors in adulthood (*Prentice and Viner, 2013*). However, childhood obesity can lead to early onset of puberty, especially among girls (*Li et al., 2017*; *Richardson et al., 2020*), and, therefore, can confound the observed associations between early pubertal development and worse later health. Moreover, early pubertal development is linked to substance use and other risky behaviors in adolescence (*Hartman et al., 2017*; *Savage et al., 2018*), but the associations are partly explained by familial factors (*Savage et al., 2018*).

Many unhealthy lifestyle choices, such as smoking initiation, alcohol use, and a physically inactive lifestyle, are already made in adolescence and increase the risk of developing several noncommunicable diseases over the following decades (*Lopez et al., 2006*). Once initiated, unhealthy habits are likely to persist into adulthood (*Latvala et al., 2014*; *Maggs and Schulenberg, 2005*; *Rovio et al., 2018*; *Salin et al., 2019*). A recent systematic review showed that healthy habits tend to cluster during childhood and adolescence, and typically, about half of the adolescents fall into subgroups characterized by healthy lifestyle habits (*Whitaker et al., 2021*). However, small minorities of adolescents are classified as heavy substance users or as having multiple other risk behaviors (*Whitaker et al., 2021*). The long-term consequences of the accumulation of unhealthy adolescent behaviors on health in later life have been rarely studied.

An unhealthy lifestyle in adolescence can affect biological mechanisms of aging at the molecular level and, subsequently, morbidity. Epigenetic alterations, including age-related changes in DNA methylation (DNAm), constitute a primary hallmark of biological aging (*López-Otín et al., 2013*). Epigenetic clocks are algorithms that aim to quantify biological aging using DNAm levels within specific CpG sites. The first-generation clocks, Horvath's and Hannum's clocks, were trained to predict chronological age (*Hannum et al., 2013*; *Horvath, 2013*), whereas the second-generation clocks, such as DNAm PhenoAge and GrimAge, are better predictors of health span and lifespan (*Levine et al., 2018*; *Lu et al., 2019*). For epigenetic clocks, the difference between an individual's epigenetic age estimate and chronological age provides a measure of age acceleration (AA). The DunedinPoAm estimator differs from its predecessors in that it has been developed to predict the pace of aging (*Belsky et al., 2020*). The pace of aging describes longitudinal changes over 12 years in several biomarkers of organ-system integrity among same-aged individuals. Recently, the DunedinPACE estimator, which constitutes an advance on the original DunedinPoAm, was published (*Belsky et al., 2022*). DunedinPACE was trained to predict pace of aging measured over 20-year follow-up, and only the reliable probes were used in the prediction. From the life-course perspective, epigenetic aging measures are useful tools to assess biological aging at all ages and detect changes induced by lifetime exposures.

Previous studies have linked several lifestyle-related factors, such as higher body mass index (BMI), smoking, alcohol use, and lower leisure-time physical activity (LTPA), with accelerated biological aging measured using epigenetic clocks (*Oblak et al., 2021*; *Quach et al., 2017*). However, most of these studies were based on cross-sectional data on older adults. The first studies on the associations of adolescent lifestyle-related exposures with biological aging assessed with epigenetic aging measures indicated that advanced pubertal development, higher BMI, and smoking are associated with accelerated biological aging in adolescence (*Etzel et al., 2022*; *Raffington et al., 2021*; *Simpkin et al., 2017*).

**eLife digest** For most animals, events that occur early in life can have a lasting impact on individuals' health. In humans, adolescence is a particularly vulnerable time when rapid growth and development collide with growing independence and experimentation. An unhealthy lifestyle during this period of rapid cell growth can contribute to later health problems like heart disease, lung disease, and premature death. This is due partly to accelerated biological aging, where the body deteriorates faster than what would be expected for an individual's chronological age.

One way to track the effects of lifestyle on biological aging is by measuring epigenetic changes. Epigenetic changes consist on adding or removing chemical 'tags' on genes. These tags can switch the genes on or off without changing their sequences. Scientists can measure certain epigenetic changes by measuring the levels of methylated DNA – DNA with a chemical 'tag' known as a methyl group – in blood samples. Several algorithms – known as 'epigenetic clocks' – are available that estimate how fast an individual is aging biologically based on DNA methylation.

Kankaanpää et al. show that unhealthy lifestyles during adolescence may lead to accelerated aging in early adulthood. For their analysis, Kankaanpää et al. used data on the levels of DNA methylation in blood samples from 824 twins between 21 and 25 years old. The twins were participants in the FinnTwin12 study and had completed a survey about their lifestyles at ages 12, 14, and 17.

Kankaanpää et al. classified individuals into five groups depending on their lifestyles. The first three groups, which included most of the twins, contained individuals that led relatively healthy lives. The fourth group contained individuals with a higher body mass index based on their height and weight. Finally, the last group included individuals with unhealthy lifestyles who binge drank, smoked and did not exercise.

After estimating the biological ages for all of the participants, Kankaanpää et al. found that both the individuals with higher body mass indices and those in the group with unhealthy lifestyles aged faster than those who reported healthier lifestyles. However, the results varied depending on which epigenetic clock Kankaanpää et al. used to measure biological aging: clocks that had been developed earlier showed fewer differences in aging between groups; while newer clocks consistently found that individuals in the higher body mass index and unhealthy groups were older. Kankaanpää et al. also showed that shared genetic factors explained both unhealthy lifestyles and accelerated biological aging.

The experiments performed by Kankaanpää et al. provide new insights into the vital role of an individual's genetics in unhealthy lifestyles and cellular aging. These insights might help scientists identify at risk individuals early in life and try to prevent accelerated aging.

The few previous studies conducted on this topic have focused on single lifestyle factors, and a comprehensive understanding of the role of adolescent lifestyle in later biological aging remains unclear. Our first aim is to define the types of lifestyle behavior patterns that can be identified in adolescence using data-driven latent class analysis (LCA). The second aim is to investigate whether the identified behavioral subgroups differ in biological aging in young adulthood and whether the associations are independent of baseline pubertal development. The third aim is to assess the genetic and environmental influences shared between biological aging and adolescent lifestyle behavior patterns.

## Methods

The participants were Finnish twins and members of the longitudinal FinnTwin12 study (born during 1983–1987) (*Kaprio, 2013*; *Rose et al., 2019*). A total of 5600 twins and their families initially enrolled in the study. At the baseline, the twins filled out the questionnaires regarding their lifestyle-related habits at 11–12 years of age, and follow-up assessments were conducted at ages 14 and 17.5 years. The response rates were high for each assessment (85–90%). In young adulthood, at an average age of 22 years, blood samples for DNA analyses were collected during in-person clinical studies after written informed consent was signed. The data on health-related behaviors were collected with questionnaires and interviews. A total of 1295 twins of the FinnTwin12 cohort were examined and measured either in-person or through telephonic interviews. DNAm was determined and biological

aging was assessed for 847 twins, out of which 824 twins had also information on lifestyle-related habits in adolescence. Data collection was conducted in accordance with the Declaration of Helsinki. The Indiana University IRB and the ethics committees of the University of Helsinki and Helsinki University Central Hospital approved the study protocol (113/E3/2001and 346/E0/05).

## DNAm and assessment of biological age

Genomic DNA was extracted from peripheral blood samples using commercial kits. High molecular weight DNA samples (1 µg) were bisulfite converted using EZ-96 DNA methylation-Gold Kit (Zymo Research, Irvine, CA) according to the manufacturer's protocol. The twins and co-twins were randomly distributed across plates, with both twins from a pair on the same plate. DNAm profiles were obtained using Illumina's Infinium HumanMethylation450 BeadChip or the Infinium MethylationEPIC BeadChip (Illumina, San Diego, CA). The Illumina BeadChips measure single-CpG resolution DNAm levels across the human genome. With these assays, it is possible to interrogate over 450,000 (450k) or 850,000 (EPIC) methylation sites quantitatively across the genome at single-nucleotide resolution. Of the samples included in this study, 744 were assayed using 450k and 80 samples using EPIC arrays. Methylation data from different platforms was combined and preprocessed together using R package minfi (*Aryee et al., 2014*). We calculated detection p-values comparing total signal for each probe to the background signal level to evaluate the quality of the samples (*Maksimovic et al., 2016*). Samples of poor quality (mean detection p>0.01) were excluded from further analysis. Data were normalized by using the single-sample Noob normalization method, which is suitable for datasets originating from different platforms (*Fortin et al., 2017*). We also used Beta-Mixture Quantile (BMIQ) normalization (*Teschendorff et al., 2013*). Beta values representing CpG methylation levels were calculated as the ratio of methylated intensities (M) to the overall intensities (beta value = M/(M + U + 100), where U is unmethylated probe intensity). These preprocessed beta values were used as input in the calculations of the estimates of epigenetic aging.

We utilized six epigenetic clocks. The first four clocks, namely, Horvath's and Hannum's epigenetic clocks (*Hannum et al., 2013*; *Horvath, 2013*) and DNAm PhenoAge and DNAm GrimAge estimators (*Levine et al., 2018*; *Lu et al., 2019*), produced DNAm-based epigenetic age estimates in years by using a publicly available online calculator (https://dnamage.genetics.ucla.edu/new) (normalization method implemented in the calculator was utilized, as well). For these measures, AA was defined as the residual obtained from regressing the estimated epigenetic age on chronological age (AA$_{Horvath}$, AA$_{Hannum}$, AA$_{Pheno}$, and AA$_{Grim}$, respectively). The fifth and sixth clocks, namely, DunedinPoAm and DunedinPACE estimators, provided an estimate for the pace of biological aging in years per calendar year (*Belsky et al., 2020*; *Belsky et al., 2022*). DunedinPoAm and DunedinPACE were calculated using publicly available R packages (https://github.com/danbelsky/DunedinPoAm38; *Belsky et al., 2020* and https://github.com/danbelsky/DunedinPACE; *Belsky et al., 2022*, respectively). The epigenetic aging measures were screened for outliers (>5 standard deviations away from mean). One outlier was detected according to DunedinPACE and was recoded as a missing value.

The components of DNAm GrimAge (adjusted for age) were also obtained, including DNAm-based smoking pack-years and the surrogates for plasma proteins (DNAm-based plasma proteins): DNAm adrenomedullin (ADM), DNAm beta-2-microglobulin (B2M), DNAm cystatin C, DNAm growth differentiation factor 15 (GDF15), DNAm leptin, DNAm plasminogen activator inhibitor 1 (PAI-1), and DNAm tissue inhibitor metalloproteinases 1 (TIMP-1).

## Lifestyle-related factors in adolescence

### BMI at ages 12, 14, and 17 years

BMI (kg/m$^2$) was calculated based on self-reported height and weight.

### LTPA at ages 12, 14, and 17 years

The frequency of LTPA at the age of 12 years was assessed with the question 'How often do you engage in sports (i.e., team sports and training)?' The answers were classified as 0 = less than once a week, 1 = once a week, and 2 = every day. At ages 14 and 17 years, the question differed slightly: 'How often do you engage in physical activity or sports during your leisure time (excluding physical education)?' The answers were classified as 0 = less than once a week, 1 = once a week, 2 = 2–5 times a week, and 3 = every day.

### Smoking status at ages 14 and 17 years

Smoking status was determined using the self-reported frequency of smoking and classified as 0 = never smoker, 1 = former smoker, 2 = occasional smoker, and 3 = daily smoker.

### Alcohol use (binge drinking) at ages 14 and 17 years

The frequency of drinking to intoxication had the following classes: 'How often do you get really drunk?' 0 = never, 1 = less than once a month, 2 = approximately once or twice a month, and 3 = once a week or more.

### Pubertal development at age 12 years

Baseline pubertal development was assessed using a modified five-item Pubertal Development Scale (PDS) questionnaire (*Petersen et al., 1988*; *Wehkalampi et al., 2008*). Both sexes answered three questions each concerning growth in height, body hair, and skin changes. Moreover, boys were asked questions about the development of facial hair and voice change, while girls were asked about breast development and menarche. Each question had response categories 1 = growth/change has not begun, 2 = growth/change has barely started, and 3 = growth/change is definitely underway, except for menarche, which was dichotomous, 1 = has not occurred or 3 = has occurred (see also *Wehkalampi et al., 2008*). PDS was calculated as the mean score of the five items, and higher values indicated more advanced pubertal development at age 12 years.

## Lifestyle-related factors in young adulthood at age 21–25 years

*BMI* ($kg/m^2$) was calculated based on the measured height and weight.

*LTPA* was assessed using the Baecke questionnaire (*Baecke et al., 1982*). A sport index was based on the mean scores of four questions on sports activity described by *Baecke et al., 1982* and *Mustelin et al., 2012* for the FinnTwin12 study. The sport index is a reliable and valid instrument to measure high-intensity physical activity (*Richardson et al., 1995*).

*Smoking* was self-reported and classified as never, former, or current smoker.

*Alcohol use* (100% alcohol grams/day) was derived from the Semi-Structured Assessment for the Genetics of Alcoholism (*Bucholz et al., 1994*) and based on quantity and frequency of use and the content of alcoholic beverages, assessed by trained interviewers.

## Statistical analysis

### Patterns of lifestyle behaviors in adolescence

To identify the patterns of lifestyle behaviors in adolescence, an LCA was conducted, which is a data-driven approach to identify homogenous subgroups in a heterogeneous population. The classification was based on BMI and LTPA at ages 12, 14, and 17 years and smoking status and alcohol use at ages 14 and 17 years (10 indicator variables). All variables were treated as ordinal variables, except for continuous BMI. The classification was based on the thresholds of the ordinal variables and the means and variances of BMI.

An LCA model with 1–8 classes was fitted. The following fit indices were used to evaluate the goodness of fit: Akaike's information criterion, Bayesian information criterion (BIC), and sample size-adjusted BIC. The lower values of the information criteria indicated a better fit for the model. Moreover, we used the Vuong–Lo–Mendell–Rubin likelihood ratio (VLMR) test and the Lo–Mendell–Rubin (LMR) test to determine the optimal number of classes. The estimated model was compared with the model with one class less, and the low p-value suggested that the model with one class less should be rejected. At each step, the classification quality was assessed using the average posterior probabilities for most likely latent class membership (AvePP). AvePP values close to 1 indicate a clear classification. In addition to the model fit, the final model for further analyses was chosen based on the parsimony and interpretability of the classes.

### Differences in biological aging

The mean differences in biological aging between the lifestyle behavior patterns were studied using the Bolck–Croon–Hagenaars approach (*Asparouhov and Muthén, 2021*). The class-specific weights for each participant were computed and saved during the latent class model estimation. After that,

a secondary model conditional on the latent lifestyle behavior patterns was specified using weights as training data: Epigenetic aging measures were treated as distal outcome one at a time, and the mean differences across classes were studied while adjusting for sex, age and baseline pubertal development. Similarly, the mean differences in the components of DNAm GrimAge and lifestyle-related factors in young adulthood were studied. The models of epigenetic aging measures were additionally adjusted for BMI in adulthood. To evaluate the effect sizes, standardized mean differences (SMDs) were calculated.

## Genetic and environmental influences

Genetic and environmental influences on biological aging in common with lifestyle behavior patterns were studied using quantitative genetic modeling. For simplicity, we adjusted the epigenetic aging variables for sex, age, and baseline pubertal development prior to the analysis.

We first carried out univariate modeling to study genetic and environmental influences on epigenetic aging measures (*Neale and Cardon, 1992*). The variance in the epigenetic aging measures was decomposed into the latent variables representing additive genetic (A), dominant genetic (D), or shared environmental (C) and non-shared environmental (E) components (ACE model or ADE model). The sequences of the models were fitted (ACE, ADE, AE, CE, and E). Because dominance in the absence of additive effects is rare, the model including D and E components (DE-model) was omitted. We used Satorra–Bentler scaled chi-squared ($\chi^2$) test, comparative fit index (CFI), Tucker–Lewis index (TLI), root mean square error of approximation (RMSEA), and standardized root-mean-square residual (SRMR) to evaluate the goodness of fit of the models. The model fits the data well if the $\chi^2$ test is not statistically significant (p>0.05), CFI and TLI values are close to 0.95, the RMSEA value is below 0.06, and the SRMR value is below 0.08 (*Hu and Bentler, 1999*). Moreover, BIC was used to compare non-nested models. A lower BIC value indicates a better model fit. The most parsimonious model with the sufficient fit to the data was considered optimal.

On the one hand, as described above, total variance in biological aging was decomposed in the components explained by genetic, shared, and unshared environmental factors $\left( a_{Tot}^2 + c_{Tot}^2 + e_{Tot}^2 \right) (= Var_{Tot})$ (*Figure 1A*). On the other hand, we can use the secondary model to study the differences in biological

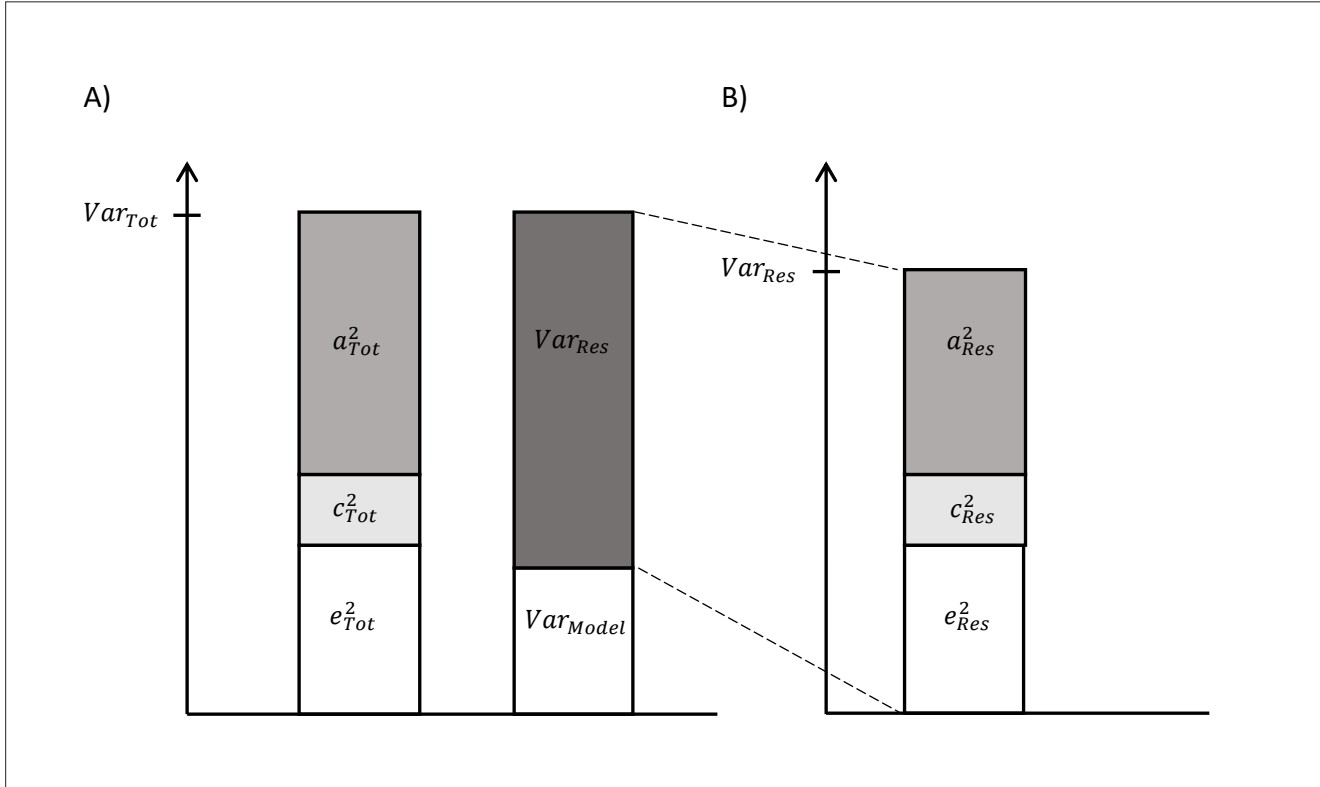

**Figure 1.** Decomposition of (**A**) total variation in biological aging and (**B**) the variation of the residual term.

aging between the adolescent lifestyle behavior patterns, as described above, and decompose the variance in biological aging into the variance explained by the adolescent lifestyle behavior patterns $Var_{(Model)}$ and the variance of the residual term $Var_{(Res)}$. We also conducted univariate modeling for the residual term of biological aging, which corresponds to the variation in biological aging not explained by the adolescent lifestyle behavior patterns $\left( a_{Res}^2 + c_{Res}^2 + e_{Res}^2 \right) (= Var_{Res})$ (**Figure 1B**). The residual terms were obtained by specifying a latent variable corresponding to the residuals of the secondary model described above (without including covariates), and the factor scores were saved. Finally, the proportion of variation in biological aging explained by the genetic factors shared with adolescent lifestyle patterns was evaluated as follows: $\left( a_{Tot}^2 - a_{Res}^2 \right) / Var_{Tot}$. The proportion of variation in epigenetic aging explained by the environmental factors was evaluated similarly. These proportions reflect the extent to which the same genetic/environmental factors contribute to the association between the adolescent lifestyle patterns and biological aging (i.e., size of the genetic and environmental correlations between the phenotypes).

Missing data were assumed to be missing at random (MAR). The model parameters were estimated using the full information maximum likelihood (FIML) method with robust standard errors. Under the MAR assumption, the FIML method produced unbiased parameter estimates. The standard errors of the latent class models and secondary models were corrected for nested sampling (TYPE = COMPLEX). Descriptive statistics were calculated using IBM SPSS Statistics for Windows, version 20.0 (IBM Corp, Armonk, NY), and further modeling was conducted using Mplus, version 8.2 (**Muthén and Muthén, 1998**).

## Results

The descriptive statistics of the study variables are presented in **Table 1**. A total of 5114 twins answered questionnaires on lifestyle-related behaviors during their adolescent years at least once. For 824 twins, epigenetic aging estimates were obtained. The mean age (SD) of the twins having information on biological aging was 22.4 (0.7) years. The means of the epigenetic age estimates were estimated as follows: Horvath's clock 28.9 (3.6), Hannum's clock 18.2 (3.3), DNAm PhenoAge 13.0 (5.3), and DNAm GrimAge 25.2 (3.3) years. The intraclass correlation coefficients (ICCs) of epigenetic aging measures were consistently higher in MZ twin pairs than in DZ twin pairs (**Table 2**). This suggests an underlying genetic component in biological aging. The correlations between the different epigenetic aging measures ranged from –0.12 to 0.73. The lowest correlation was observed between $AA_{Horvath}$ and DunedinPoAm and between $AA_{Horvath}$ and DunedinPACE. All other correlations were positive. The highest correlations (>0.5) were observed between $AA_{Hannum}$ and $AA_{Pheno}$, $AA_{Grim}$ and DunedinPoAm, and DunedinPoAm and DunedinPACE.

### Patterns of lifestyle behaviors

Increasing the number of classes continued to improve AIC, BIC, and ABIC (**Table 3**). However, the VLMR and LMR tests indicated that even a solution with four classes would be sufficient. In the fifth step, a class of participants with high BMI was extracted. Previous studies have shown the role of being overweight or obese in biological aging (**Lundgren et al., 2022**). After including the sixth class, the information criteria still showed considerable improvement, but the AvePPs for several classes were below 0.8. For these reasons, and to have adequate statistical power for subsequent analyses, a five-class solution was considered optimal. The AvePPs ranged from 0.78 to 0.91 for the five-class solution, indicating reasonable classification quality.

Of the participants, 32% fell into the class of healthiest lifestyle habits (C1) (see **Figure 2**, and the distributions of indicator variables according to the adolescent lifestyle behavior patterns in **Table 4**). They had normal weight, on average, and were more likely to engage in regular LTPA compared to the other groups; most of them were non-smokers and did not use alcohol regularly. Every fifth (19.9%) participant belonged to the second class (C2), characterized by the low mean level of BMI in the range of normal weight for children (low-normal BMI) (**Cole et al., 2007**). They also had healthy lifestyle habits, but they were not as physically active as the participants in class C1. The participants placed in the third class (C3, 22.8%) had lifestyle habits similar to those of the participants in class C1; however, they had a higher level of BMI in the range of normal weight for children (high-normal BMI). About every tenth (9.5%) of the participants belonged to the fourth class (C4), with the highest

**Table 1.** Descriptive statistics of the adolescent lifestyle-related variables in all twins and in the subsample of twins with information on biological aging.

| | All twins (n = 5114) | | Subsample (n = 824) | |
| --- | --- | --- | --- | --- |
| | n | Mean (SD) or % | n | Mean (SD) or % |
| Zygosity | 4852 | | 824 | |
| MZ | 1650 | 34.0 | 335 | 40.7 |
| Same-sex DZ | 1603 | 33.0 | 262 | 31.8 |
| Opposite-sex DZ | 1599 | 33.0 | 227 | 27.5 |
| Sex | 5114 | | 824 | |
| Female | 2584 | 50.5 | 470 | 57.0 |
| Male | 2530 | 49.5 | 354 | 43.0 |
| At age 12 | | | | |
| Pubertal development (1–3) | 5111 | 1.6 (0.5) | 823 | 1.6 (0.5) |
| Body mass index | 4913 | 17.6 (2.6) | 793 | 17.7 (2.6) |
| Leisure-time physical activity | 5038 | | 813 | |
| Less than once a week | 1877 | 37.3 | 295 | 35.3 |
| Once a week | 2499 | 49.6 | 416 | 51.2 |
| Every day | 662 | 13.1 | 102 | 12.5 |
| At age 14 | | | | |
| Body mass index | 4473 | 19.3 (2.7) | 787 | 19.5 (2.6) |
| Leisure-time physical activity | 4590 | | 799 | |
| Less than once a week | 688 | 15.0 | 110 | 13.8 |
| Once a week | 796 | 17.3 | 149 | 18.6 |
| 2–5 times a week | 2182 | 47.5 | 370 | 46.3 |
| Every day | 924 | 20.1 | 170 | 21.3 |
| Smoking status | 4570 | | 800 | |
| Never | 3954 | 86.5 | 687 | 85.9 |
| Former | 296 | 6.5 | 57 | 7.1 |
| Occasional | 122 | 2.7 | 24 | 3.0 |
| Daily smoker | 198 | 4.3 | 32 | 4.0 |
| Alcohol use (binge drinking) | 4565 | | 796 | |
| Never | 3501 | 76.7 | 602 | 75.6 |
| Less than once a month | 756 | 16.6 | 135 | 17 |
| Once or twice a month | 275 | 6.0 | 50 | 6.3 |
| Once a week or more | 33 | 0.7 | 9 | 1.1 |
| At age 17 | | | | |
| Body mass index | 4158 | 21.4 (3.0) | 760 | 21.4 (2.7) |
| Leisure-time physical activity | 4208 | | 766 | |
| Less than once a week | 748 | 17.8 | 132 | 17.2 |
| Once a week | 686 | 16.3 | 130 | 17.0 |
| 2–5 times a week | 1977 | 47.0 | 363 | 47.4 |

*Table 1 continued on next page*

*Table 1 continued*

| | All twins (n = 5114) | | Subsample (n = 824) | |
|---|---|---|---|---|
| | n | Mean (SD) or % | n | Mean (SD) or % |
| Every day | 797 | 18.9 | 141 | 18.4 |
| Smoking status | 4190 | | 762 | |
| Never | 2419 | 57.7 | 454 | 59.7 |
| Former | 493 | 11.8 | 83 | 10.9 |
| Occasional | 213 | 5.1 | 48 | 6.3 |
| Daily smoker | 1065 | 25.4 | 176 | 23.1 |
| Alcohol use (binge drinking) | 4217 | | 766 | |
| Never | 881 | 20.9 | 152 | 19.8 |
| Less than once a month | 1807 | 42.9 | 340 | 44.4 |
| Once or twice a month | 1240 | 29.4 | 222 | 29.0 |
| Once a week or more | 289 | 6.9 | 52 | 6.8 |

MZ, monozygotic twins; DZ, dizygotic twins; SD, standard deviation.

level of BMI (high BMI). At each measurement point, the mean BMI level exceeded the cutoff points for overweight in children (*Cole et al., 2007*). The prevalence of daily smoking was slightly higher in C4 compared to classes C1, C2, and C3. Of the participants, 15.9% were classified into the subgroup characterized by the unhealthiest lifestyle behaviors (C5). Most of them were daily smokers and used alcohol regularly at the age of 17. They also had a lower probability of engaging in regular LTPA compared to the other groups; however, they were of normal weight, on average.

Boys were slightly over-represented in the classes that were most physically active (C1, C3) and had the highest levels of BMI (C3, C4) (percentage of boys: C1: 57.2%; C3: 51.5%; and C4: 52.7%), and under-represented in the classes with lowest levels of BMI (C2) and the unhealthiest lifestyle behavior pattern (C5) (C2: 42.7%; C5: 44.1%). There were also differences in pubertal development at baseline between the groups. The subgroups with the highest levels of BMI (C3, C4) and the class with

**Table 2.** The intraclass correlation coefficients (ICCs) of epigenetic aging measures by zygosity and correlation coefficients between the measures (n = 824).

| | ICCs (95% CI) | | Correlation coefficients (95% CI) off-diagonal and means (standard deviations) on the diagonal | | | | | |
|---|---|---|---|---|---|---|---|---|
| | MZ twin pairs | DZ twin pairs | $AA_{Horvath}$ | $AA_{Hannum}$ | $AA_{Pheno}$ | $AA_{Grim}$ | DunedinPoAm | DunedinPACE |
| $AA_{Horvath}$ | 0.71 (0.63, 0.79) | 0.40 (0.24, 0.55) | 0.00 (3.51) | | | | | |
| $AA_{Hannum}$ | 0.66 (0.56, 0.76) | 0.32 (0.16, 0.48) | 0.40 (0.33, 0.48) | 0.00 (3.27) | | | | |
| $AA_{Pheno}$ | 0.69 (0.60, 0.78) | 0.16 (0.00, 0.33) | 0.36 (0.29, 0.44) | 0.61 (0.56, 0.66) | 0.00 (5.25) | | | |
| $AA_{Grim}$ | 0.72 (0.63, 0.80) | 0.35 (0.15, 0.55) | 0.08 (0.01, 0.16) | 0.32 (0.24, 0.40) | 0.39 (0.33, 0.46) | 0.00 (3.24) | | |
| DunedinPoAm | 0.62 (0.52, 0.71) | 0.42 (0.24, 0.60) | −0.05 (-0.12, 0.03) | 0.20 (0.13, 0.27) | 0.41 (0.35, 0.47) | 0.57 (0.52, 0.63) | 1.00 (0.07) | |
| DunedinPACE | 0.71 (0.64, 0.78) | 0.46 (0.31, 0.61) | −0.04 (−0.11, 0.04) | 0.30 (0.22, 0.38) | 0.49 (0.43, 0.55) | 0.55 (0.49, 0.61) | 0.62 (0.57, 0.67) | 0.88 (0.10) |

CIs were corrected for nested sampling.
CI, confidence interval; AA, age acceleration; MZ, monozygotic; DZ, dizygotic.

**Table 3.** Model fit of the latent class models (n = 5114).

| AIC | BIC | ABIC | VLMR | LMR | Class sizes | AvePP |
|---|---|---|---|---|---|---|
| 128842 | 129012 | 128929 | | | | |
| 122533 | 122880 | 122711 | <0.001 | <0.001 | 74.0%, 26.0% | 0.95, 0.92 |
| 119937 | 120460 | 120206 | <0.001 | <0.001 | 44.9%, 40.5%, 14.6% | 0.88, 0.89, 0.93 |
| 118030 | 118729 | 118389 | <0.001 | <0.001 | 36.4%, 32.7%, 16.7%, 14.2% | 0.83, 0.86, 0.87, 0.92 |
| 117167 | 118043 | 117617 | 0.529 | 0.530 | 32.0%, 22.8%, 19.9%, 15.9%, 9.5% | 0.78, 0.82, 0.85, 0.88, 0.91 |
| 116526 | 117578 | 117076 | 0.169 | 0.170 | 31.5%, 18.5%, 15.7%, 14.0%, 12.7%, 7.7% | 0.77, 0.84, 0.83, 0.78, 0.78, 0.90 |
| 116099 | 117328 | 116731 | 0.043 | 0.044 | 21.0%, 17.5%, 15.2%, 13.8%, 12.9%, 12.8%, 6.9% | 0.73, 0.82, 0.70, 0.77, 0.83, 0.83, 0.91 |
| 115695 | 117101 | 116418 | 0.407 | 0.408 | 20.3%, 16.2%, 13.6%, 13.5%, 12.3%, 11.3%, 9.3%, 3.4% | 0.72, 0.75, 0.82, 0.71, 0.83, 0.80, 0.82, 0.89 |

AIC, Akaike's information criterion; BIC, Bayesian information criterion; ABIC, sample size-adjusted Bayesian information criterion; VLMR, Vuong–Lo–Mendell–Rubin likelihood ratio test; LMR, Lo–Mendell–Rubin-adjusted likelihood ratio test; AvePP, average posterior probabilities for most likely latent class membership.

unhealthiest lifestyle habits (C5) were, on average, the most advanced in pubertal development (mean PDS, C3: 1.67 95% CI: [1.63–1.71], C4: 1.69 [1.64–1.74]; and C5: 1.68 [1.63–1.72]), while the class with the healthiest lifestyle pattern (C1) and that with the lowest level of BMI (C2) were less advanced in pubertal development (C1: 1.53 [1.50–1.56]; C2: 1.44 [1.41–1.47]).

The distribution of lifestyle behavior patterns in the subsample of participants having information on biological aging was very similar to that in the large cohort data (C1: 33.0%; C2: 16.6%; C3: 20.6%; C4: 10.1%; C5: 19.7%). In the subsample, the differences in lifestyle-related factors were maintained well over the transition from adolescence to young adulthood (*Figure 2—figure supplement 1*).

## Differences in biological aging

There were differences among the classes in AA$_{Pheno}$ (Wald test: p=0.006), AA$_{Grim}$ (p=2.3e-11), DunedinPoAm (p=3.1e-9), and DunedinPACE (p=5.5e-7) in the models adjusted for sex, age, and baseline pubertal development. There were no differences in biological aging when Horvath's clock (p=0.550) and Hannum's clock (p=0.487) were used. The overall results considering AA$_{Grim}$, DunedinPoAm, and DunedinPACE were very similar (*Figure 3* and *Table 5*).

The group with the unhealthiest lifestyle pattern (C5) was, on average, 1.7–3.3 years biologically older than the groups with healthier lifestyle patterns and normal weight (C1–C3) when DNAm GrimAge was used to assess biological aging (*Table 5*, M1). Moreover, the unhealthiest group had, an average, 2–3 weeks/calendar year faster pace of biological aging, as measured with DunedinPoAm. The differences in DunedinPACE were very similar to those observed in DunedinPoAm, but there was no difference between the unhealthiest class (C5) and the class with a healthy lifestyle and high-normal BMI (C3) and, moreover, the difference between the healthiest class (C1) was not significant at 0.01 level.

When DNAm GrimAge was used, the group with a high BMI (C4) was, on average, 1.8–2.4 years biologically older than the two groups with healthier lifestyle patterns (C1 and C2) (*Table 5*, M1). When measured with the DunedinPoAm estimator, the class had, on average, 3–4 weeks/calendar year faster pace of aging, and when measured with the DunedinPACE estimator, it had 4–5 weeks/calendar year faster pace of aging. Moreover, when DunedinPoAm and DunedinPACE were used, the class had approximately 3 weeks/calendar year faster pace of aging compared to the group with healthy lifestyle with normal-high BMI (C3), and when DunedinPACE was used, the class had 2 weeks/calendar year faster pace of aging compared to the group with unhealthiest lifestyle pattern (C5). When DNAm PhenoAge was used to assess biological aging, only the group with a high BMI stood out. The group was biologically 2.0–2.5 years older than the groups with lower mean levels of BMI (C1–C2, C5). Based on the estimation results of the models, baseline pubertal development was associated with advanced biological aging only when Hannum's clock was used to derive biological AA (standardized regression coefficient B = 0.10 [0.01–0.18]).

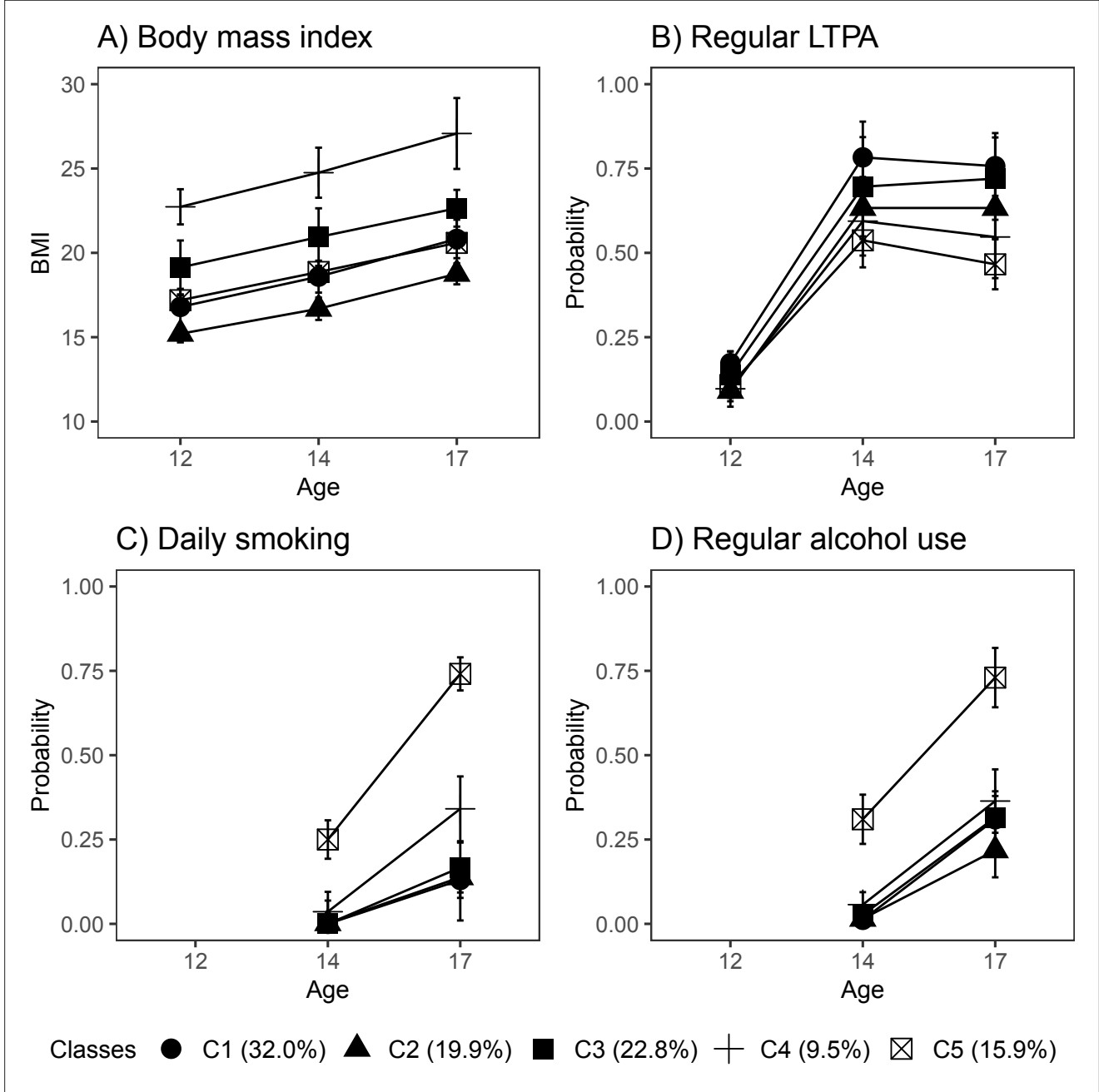

**Figure 2.** Classes with different lifestyle patterns (n = 5114). Mean and probability profiles (95% confidence intervals) of the indicator variables utilized in the classification: (**A**) body mass index, (**B**) regular leisure-time physical activity (LTPA) (several times a week), (**C**) daily smoking, and (**D**) regular alcohol use (once a month or more). For categorical variables, the probabilities of belonging to the highest categories are presented.

The online version of this article includes the following source data and figure supplement(s) for figure 2:

**Source data 1.** The estimation results of a latent class analysis (LCA) model with five classes.

**Figure supplement 1.** Lifestyle-related factors in adulthood (21–25 years) according to the adolescent lifestyle behavior classes in the subsample of participants with information on biological aging (n = 824).

**Figure supplement 1—source data 1.** Means and 95% confidence intervals of the lifestyle-related factors in adulthood according to the adolescent lifestyle behavior classes (BCH approach).

**Table 4.** The classes with different adolescent lifestyle behavior patterns (n = 5114).

| | C1 (32.0%) | | C2 (19.9%) | | C3 (22.8%) | | C4 (9.5%) | | C5 (15.9%) | |
|---|---|---|---|---|---|---|---|---|---|---|
| | Est | 95% CI | Est | 95% CI | Est | 95% CI | Est | 95% CI | Est | 95% CI |
| Body mass index | | | | | | | | | | |
| At age of 12 years | 16.8 | 15.7, 17.9 | 15.2 | 14.7, 15.7 | 19.1 | 17.5, 20.7 | 22.7 | 21.7, 23.8 | 17.2 | 16.9, 17.5 |
| At age of 14 years | 18.6 | 17.6, 19.5 | 16.7 | 16.0, 17.3 | 20.9 | 19.2, 22.6 | 24.8 | 23.3, 26.2 | 18.9 | 18.6, 19.2 |
| At age of 17 years | 20.8 | 19.7, 22.0 | 18.8 | 18.1, 19.4 | 22.6 | 21.6, 23.7 | 27.1 | 25.0, 29.2 | 20.6 | 20.3, 20.9 |
| Leisure-time physical activity | | | | | | | | | | |
| At age of 12 years | | | | | | | | | | |
| Less than once a week | 0.29 | 0.22, 0.37 | 0.45 | 0.39, 0.51 | 0.35 | 0.26, 0.43 | 0.44 | 0.37, 0.50 | 0.44 | 0.39, 0.48 |
| Once a week | 0.54 | 0.48, 0.59 | 0.46 | 0.41, 0.50 | 0.52 | 0.47, 0.56 | 0.47 | 0.39, 0.54 | 0.46 | 0.41, 0.50 |
| Every day | 0.17 | 0.14, 0.21 | 0.09 | 0.04, 0.14 | 0.14 | 0.07, 0.21 | 0.10 | 0.06, 0.13 | 0.11 | 0.08, 0.14 |
| At age of 14 years | | | | | | | | | | |
| Less than once a week | 0.08 | 0.05, 0.11 | 0.17 | 0.12, 0.22 | 0.14 | 0.07, 0.22 | 0.18 | 0.13, 0.23 | 0.27 | 0.22, 0.31 |
| Once a week | 0.14 | 0.07, 0.20 | 0.20 | 0.17, 0.24 | 0.16 | 0.10, 0.23 | 0.23 | 0.17, 0.28 | 0.20 | 0.16, 0.23 |
| 2–5 times a week | 0.52 | 0.45, 0.59 | 0.45 | 0.41, 0.49 | 0.51 | 0.43, 0.59 | 0.43 | 0.37, 0.49 | 0.40 | 0.35, 0.45 |
| Every day | 0.27 | 0.23, 0.30 | 0.18 | 0.13, 0.23 | 0.19 | 0.12, 0.25 | 0.17 | 0.12, 0.21 | 0.14 | 0.10, 0.17 |
| At age of 17 years | | | | | | | | | | |
| Less than once a week | 0.10 | 0.05, 0.14 | 0.19 | 0.14, 0.23 | 0.13 | 0.06, 0.20 | 0.27 | 0.19, 0.35 | 0.35 | 0.29, 0.40 |
| Once a week | 0.15 | 0.11, 0.18 | 0.18 | 0.15, 0.21 | 0.15 | 0.11, 0.19 | 0.18 | 0.14, 0.23 | 0.19 | 0.15, 0.23 |
| 2–5 times a week | 0.50 | 0.44, 0.56 | 0.45 | 0.41, 0.49 | 0.53 | 0.48, 0.57 | 0.44 | 0.36, 0.52 | 0.36 | 0.32, 0.41 |
| Every day | 0.26 | 0.22, 0.29 | 0.18 | 0.13, 0.23 | 0.20 | 0.12, 0.27 | 0.11 | 0.07, 0.15 | 0.10 | 0.07, 0.13 |
| Smoking status | | | | | | | | | | |
| At age of 14 years | | | | | | | | | | |
| Never | 0.99 | 0.98, 1.00 | 0.98 | 0.95, 1.00 | 0.97 | 0.95, 1.00 | 0.83 | 0.74, 0.93 | 0.33 | 0.24, 0.43 |
| Former | 0.01 | 0.00, 0.02 | 0.02 | 0.00, 0.03 | 0.02 | 0.00, 0.04 | 0.09 | 0.04, 0.14 | 0.29 | 0.24, 0.34 |
| Occasional | 0.00 | | 0.01 | –0.01, 0.02 | 0.00 | 0.00, 0.01 | 0.04 | 0.01, 0.07 | 0.13 | 0.10, 0.16 |
| Daily smoker | 0.00 | | 0.00 | 0.00, 0.01 | 0.00 | | 0.04 | 0.00, 0.07 | 0.25 | 0.19, 0.31 |
| At age of 17 years | | | | | | | | | | |
| Never | 0.69 | 0.61, 0.77 | 0.73 | 0.65, 0.81 | 0.68 | 0.59, 0.78 | 0.50 | 0.41, 0.59 | 0.03 | 0.00, 0.06 |
| Former | 0.12 | 0.09, 0.15 | 0.09 | 0.05, 0.13 | 0.12 | 0.07, 0.16 | 0.11 | 0.06, 0.16 | 0.15 | 0.12, 0.19 |
| Occasional | 0.06 | 0.04, 0.07 | 0.04 | 0.02, 0.06 | 0.04 | 0.01, 0.06 | 0.05 | 0.02, 0.07 | 0.07 | 0.05, 0.10 |
| Daily smoker | 0.13 | 0.08, 0.18 | 0.14 | 0.09, 0.18 | 0.17 | 0.09, 0.24 | 0.34 | 0.24, 0.44 | 0.74 | 0.69, 0.79 |
| Alcohol use (binge drinking) | | | | | | | | | | |
| At age of 14 years | | | | | | | | | | |
| Never | 0.88 | 0.85, 0.91 | 0.94 | 0.90, 0.97 | 0.84 | 0.79, 0.89 | 0.76 | 0.69, 0.83 | 0.23 | 0.15, 0.31 |
| Less than once a month | 0.11 | 0.08, 0.14 | 0.05 | 0.02, 0.08 | 0.13 | 0.09, 0.17 | 0.18 | 0.12, 0.24 | 0.46 | 0.41, 0.51 |
| Once or twice a month | 0.01 | 0.00, 0.02 | 0.02 | 0.00, 0.03 | 0.03 | 0.01, 0.04 | 0.05 | 0.02, 0.08 | 0.27 | 0.22, 0.32 |
| Once a week or more | 0.00 | | 0.00 | | 0.00 | | 0.00 | 0.00, 0.01 | 0.04 | 0.02, 0.06 |
| At age of 17 years | | | | | | | | | | |
| Never | 0.21 | 0.18, 0.25 | 0.33 | 0.26, 0.41 | 0.22 | 0.16, 0.28 | 0.23 | 0.15, 0.30 | 0.01 | 0.00, 0.02 |

*Table 4 continued on next page*

*Table 4 continued*

| | C1 (32.0%) | | C2 (19.9%) | | C3 (22.8%) | | C4 (9.5%) | | C5 (15.9%) | |
|---|---|---|---|---|---|---|---|---|---|---|
| | Est | 95% CI | Est | 95% CI | Est | 95% CI | Est | 95% CI | Est | 95% CI |
| Less than once a month | 0.48 | 0.43, 0.52 | 0.45 | 0.40, 0.49 | 0.46 | 0.41, 0.52 | 0.41 | 0.35, 0.47 | 0.26 | 0.22, 0.31 |
| Once or twice a month | 0.28 | 0.24, 0.32 | 0.18 | 0.12, 0.24 | 0.28 | 0.23, 0.33 | 0.29 | 0.23, 0.35 | 0.51 | 0.46, 0.55 |
| Once a week or more | 0.03 | 0.00, 0.06 | 0.04 | 0.02, 0.06 | 0.03 | 0.00, 0.06 | 0.08 | 0.04, 0.11 | 0.22 | 0.18, 0.26 |

Mean and probability profiles of the indicator variables utilized in the classification.
BMI, body mass index; Est, estimated mean or probability; CI, confidence interval; C1, the class with the healthiest lifestyle pattern; C2, the class with low-normal BMI; C3, the class with healthy lifestyle and high-normal BMI; C4, the class with high BMI; C5, the class with the unhealthiest lifestyle pattern.

According to the previous literature, it is controversial whether childhood obesity has a direct effect on later health or whether the association is fully mediated by BMI in adulthood (*Park et al., 2012*). The role of adult BMI may depend on which disease outcome is studied (*Richardson et al., 2020*). After additionally adjusting the model for BMI in adulthood, the differences in $AA_{Pheno}$ and DunedinPACE between the class of participants with high BMI (C4) and those with lower BMI (C1, C2, C5) were attenuated (*Table 5*, M2). This finding suggests that the observed differences in biological aging probably are fully mediated by BMI in adulthood. However, the differences in biological aging were only slightly attenuated when the DNAm GrimAge and DunedinPoAm estimators were used, suggesting that childhood overweight may leave permanent imprint on biological aging assessed with these measures. However, when DNAm GrimAge was used, the difference between the classes C4 and C1 was not significant at 0.01 level.

In our study, high standard deviations of epigenetic age estimates were observed. Therefore, variation in AA measures may largely be attributable to technical variation, which is not biologically meaningful. Recently developed principal component (PC)-based clocks are shown to improve the reliability and validity of epigenetic clocks (*Higgins-Chen et al., 2022*). We therefore replicated our main analyses using PC-based epigenetic clocks (data not shown). The standard deviations of epigenetic age estimates were similar or even higher compared with those obtained with the original clocks, but the correlations between AA measures assessed with different clocks were consistently higher when PC-based epigenetic clocks were used. Importantly, the observed associations with the adolescent lifestyle behavior patterns did not substantially change.

## Differences in DNAm-based plasma proteins and smoking pack-years

Overall, after controlling for sex, age, and baseline pubertal development, there were differences in DNAm-based ADM (Wald test: p=0.010), B2M (p=0.014), and Packyrs (p=1.3e-5), but not in DNAm-based cystatin C (p=0.140), GDF15 (p=0.228), Leptin (p=0.228), PAI-1 (p=0.055), and TIMP-1 (p=0.089) between the adolescent lifestyle behavior patterns. The class with the unhealthiest lifestyle habits (C5) differed unfavorably from the other classes only by DNAm smoking pack-years while the class of participants with high BMI (C4) stood out by several DNAm-based plasma proteins including DNAm ADM, PAI-1, and TIMP-1 (*Figure 3—figure supplement 1*).

## Genetic and environmental effects

Twin pairs with biological aging data on both members of the pair were used in the quantitative genetic modeling to estimate the genetic and environmental components of variance for biological aging (n = 154 monozygotic and 211 dizygotic pairs). The model including additive genetic and non-shared environmental component (AE model) was considered optimal for all the epigenetic aging measures (*Table 6*). Generally, ACE and ADE fit the data about as well, and models without genetic component (CE model) provided significantly worse fit. Based on these results, AE model was also chosen for the further modeling of the residual term of biological aging. Genetic factors explained 62–73% of the total variation in biological aging depending on the estimator. The rest of the variation (27–38%) was explained by unshared environmental factors.

The proportion of the total variation in biological aging in early adulthood explained by adolescent lifestyle behavior patterns was 3.7% for $AA_{Pheno}$, 16.8% for $AA_{Grim}$, 15.4% for DunedinPoAm,

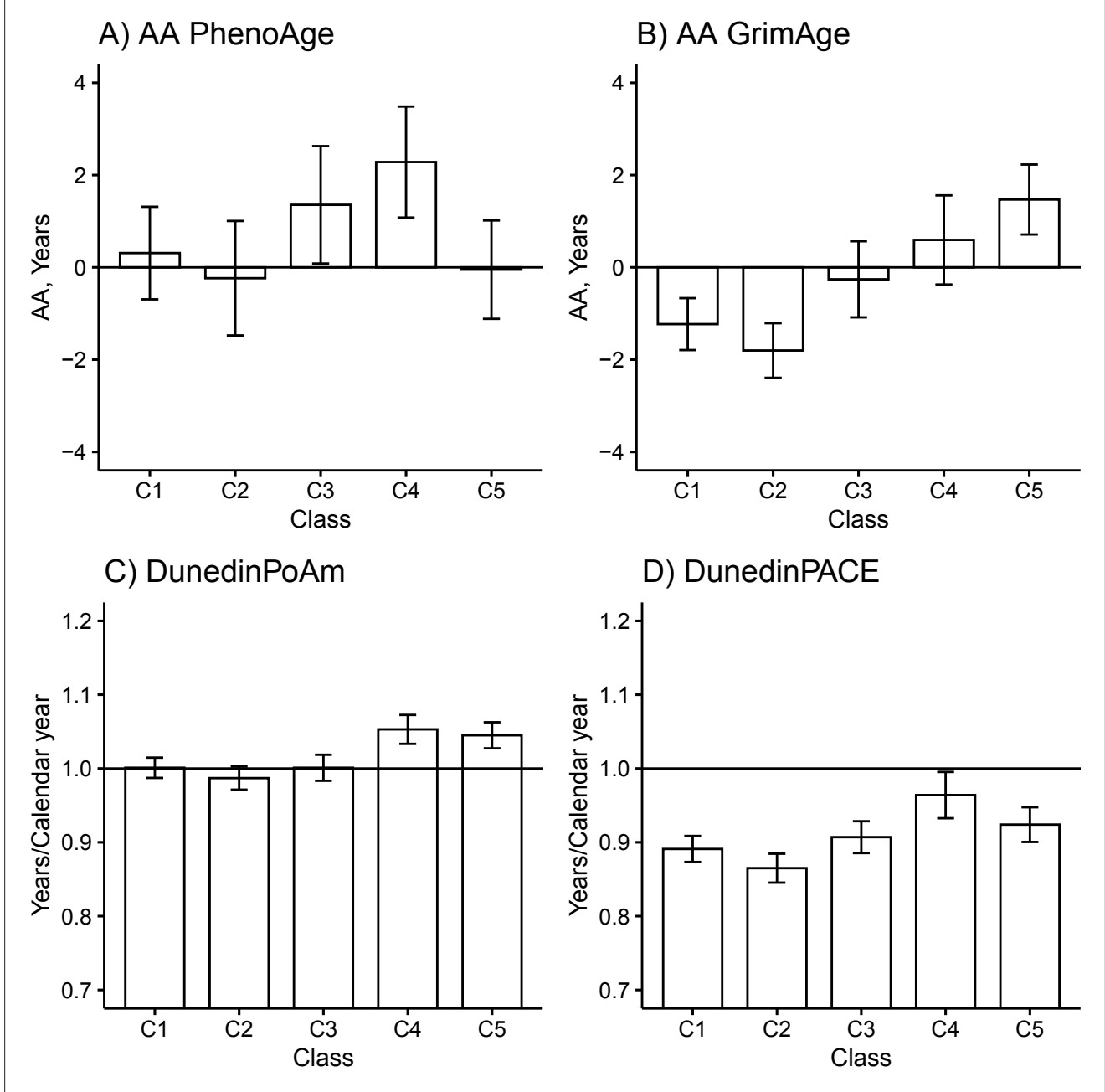

**Figure 3.** Mean differences between the adolescent lifestyle behavior patterns in biological aging measured with (**A**) DNAm PhenoAge, (**B**) DNAm GrimAge, (**C**) DunedinPoAm, and (**D**) DunedinPACE estimators (n = 824). The analysis was adjusted for sex (female), standardized age, and baseline pubertal development. Means and 95% confidence intervals are presented. C1, the class with the healthiest lifestyle pattern; C2, the class with low-normal body mass index (BMI); C3, the class with a healthy lifestyle and high-normal BMI; C4, the class with high BMI; C5, the class with the unhealthiest lifestyle pattern; AA, age acceleration.

The online version of this article includes the following source data and figure supplement(s) for figure 3:

**Source data 1.** Means and 95% confidence intervals of biological aging according to the adolescent lifestyle behavior patterns (BCH approach).

**Figure supplement 1.** DNA methylation (DNAm)-based plasma proteins and smoking pack-years according to the adolescent lifestyle behavior patterns (n = 824).

**Figure supplement 1—source data 1.** Means and 95% confidence intervals of DNA methylation (DNAm)-based plasma proteins and smoking pack-years according to the adolescent lifestyle behavior patterns (BCH approach).

**Table 5.** Differences in biological aging between classes with different adolescent lifestyle behavior patterns.

| | $AA_{Pheno}$ | | | $AA_{Grim}$ | | | DunedinPoAm | | | DunedinPACE | | |
|---|---|---|---|---|---|---|---|---|---|---|---|---|
| | Diff | 95% CI | SMD | Diff | 95% CI | SMD | Diff | 95% CI | SMD | Diff | 95% CI | SMD |
| **C2 vs. C1** | | | | | | | | | | | | |
| M1 | −0.55 | −2.15, 1.06 | −0.10 | −0.57 | −1.37, 0.23 | −0.18 | −0.01 | −0.03, 0.01 | −0.14 | −0.03 | −0.05, 0.00 | −0.30 |
| M2 | −0.13 | −1.79, 1.54 | −0.02 | −0.54 | −1.38, 0.29 | −0.17 | −0.01 | −0.03, 0.01 | −0.14 | −0.01 | −0.04, 0.02 | −0.10 |
| **C3 vs. C1** | | | | | | | | | | | | |
| M1 | 1.04 | −0.54, 2.63 | 0.20 | 0.97 | −0.01, 1.95 | 0.30 | 0.00 | −0.02, 0.02 | 0.00 | 0.02 | −0.01, 0.04 | 0.20 |
| M2 | 0.60 | −1.01, 2.21 | 0.11 | 0.94 | −0.10, 1.97 | 0.29 | 0.00 | −0.02, 0.02 | 0.00 | 0.00 | −0.03, 0.03 | 0.00 |
| **C4 vs. C1** | | | | | | | | | | | | |
| M1 | 1.97 | 0.44, 3.50 | 0.38 | 1.83 | 0.74, 2.91* | 0.56 | 0.05 | 0.03, 0.07* | 0.71 | 0.07 | 0.04, 0.11* | 0.70 |
| M2 | 0.66 | −1.31, 2.63 | 0.13 | 1.73 | 0.26, 3.21 | 0.53 | 0.04 | 0.01, 0.07* | 0.57 | 0.02 | −0.02, 0.07 | 0.20 |
| **C5 vs. C1** | | | | | | | | | | | | |
| M1 | −0.36 | −1.76, 1.04 | −0.07 | 2.70 | 1.74, 3.66* | 0.83 | 0.04 | 0.02, 0.07* | 0.57 | 0.03 | 0.00, 0.06 | 0.30 |
| M2 | −0.45 | −1.82, 0.93 | −0.09 | 2.69 | 1.73, 3.66* | 0.83 | 0.04 | 0.02, 0.06* | 0.57 | 0.03 | 0.00, 0.06 | 0.30 |
| **C3 vs. C2** | | | | | | | | | | | | |
| M1 | 1.59 | −0.07, 3.25 | 0.30 | 1.54 | 0.58, 2.50* | 0.48 | 0.01 | −0.01, 0.04 | 0.14 | 0.04 | 0.01, 0.07* | 0.50 |
| M2 | 0.73 | −1.10, 2.55 | 0.14 | 1.48 | 0.36, 2.60* | 0.46 | 0.01 | −0.02, 0.03 | 0.14 | 0.01 | −0.03, 0.04 | 0.10 |
| **C4 vs. C2** | | | | | | | | | | | | |
| M1 | 2.52 | 0.85, 4.18* | 0.48 | 2.40 | 1.28, 3.51* | 0.74 | 0.07 | 0.04, 0.09* | 1.00 | 0.10 | 0.06, 0.14* | 1.00 |
| M2 | 0.79 | −1.59, 3.16 | 0.15 | 2.27 | 0.59, 3.95* | 0.70 | 0.05 | 0.02, 0.09* | 0.71 | 0.03 | −0.02, 0.08 | 0.30 |
| **C5 vs. C2** | | | | | | | | | | | | |
| M1 | 0.19 | −1.40, 1.77 | 0.04 | 3.27 | 2.32, 4.23* | 1.01 | 0.06 | 0.03, 0.08* | 0.86 | 0.06 | 0.03, 0.09* | 0.60 |
| M2 | −0.32 | −1.97, 1.33 | −0.06 | 3.24 | 2.21, 4.27* | 1.00 | 0.05 | 0.03, 0.08* | 0.71 | 0.04 | 0.01, 0.07 | 0.40 |
| **C4 vs. C3** | | | | | | | | | | | | |
| M1 | 0.93 | −0.82, 2.67 | 0.18 | 0.85 | −0.45, 2.16 | 0.26 | 0.05 | 0.03, 0.08* | 0.71 | 0.06 | 0.02, 0.10* | 0.60 |
| M2 | 0.06 | −1.91, 2.03 | 0.01 | 0.79 | −0.68, 2.26 | 0.24 | 0.05 | 0.02, 0.08* | 0.71 | 0.02 | −0.02, 0.07 | 0.20 |
| **C5 vs. C3** | | | | | | | | | | | | |
| M1 | −1.40 | −2.99, 0.18 | −0.27 | 1.73 | 0.62, 2.84* | 0.53 | 0.04 | 0.02, 0.07* | 0.57 | 0.02 | −0.02, 0.05 | 0.20 |
| M2 | −1.05 | −2.63, 0.54 | −0.20 | 1.76 | 0.63, 2.88* | 0.54 | 0.05 | 0.02, 0.07* | 0.71 | 0.03 | 0.00, 0.06 | 0.30 |
| **C5 vs. C4** | | | | | | | | | | | | |

*Table 5 continued on next page*

*Table 5 continued*

| | AA$_{Pheno}$ | | | AA$_{Grim}$ | | | DunedinPoAm | | | DunedinPACE | | |
|---|---|---|---|---|---|---|---|---|---|---|---|---|
| | **Diff** | **95% CI** | **SMD** | **Diff** | **95% CI** | **SMD** | **Diff** | **95% CI** | **SMD** | **Diff** | **95% CI** | **SMD** |
| M1 | −2.33 | −3.84, −0.82* | −0.44 | 0.88 | −0.32, 2.07 | 0.27 | −0.01 | −0.03, 0.02 | −0.14 | −0.04 | −0.08, 0.00 | −0.40 |
| M2 | −1.10 | −3.01, 0.80 | −0.21 | 0.96 | −0.51, 2.44 | 0.30 | 0.00 | −0.03, 0.03 | 0.00 | 0.01 | −0.04, 0.05 | 0.10 |

AA, age acceleration; BMI, body mass index; Diff, difference; CI, confidence interval; SMD, standardized mean difference; C1, the class with the healthiest lifestyle pattern; C2, the class with low-normal BMI; C3, the class with healthy lifestyle and high-normal BMI; C4, the class with high BMI; C5, the class with the unhealthiest lifestyle pattern; M1, model was adjusted for sex, age, and pubertal status at age 12; M2, model was additionally adjusted for BMI in adulthood.

*The corresponding 99% confidence interval did not overlap zero.

and 10.5% for DunedinPACE (*Figure 4*). The association between adolescent lifestyle patterns and biological aging in early adulthood was largely explained by shared genetic influences; the genetic factors shared with adolescent lifestyle explained 3.7, 13.1, 12.6, and 10.5%, respectively, of the total variation in biological aging. Depending on the biological aging estimate, only 0–3.7% of the total variation in biological aging was explained by (unshared) environmental factors shared with adolescent lifestyle patterns. The rest of the total variation in biological aging was explained by genetic and (unshared) environmental factors unique to biological aging.

## Discussion

We conducted a twin study with a longitudinal lifestyle follow-up during the adolescent years and measured biological aging from genome-wide DNAm data using the most recent epigenetic aging clocks. Our findings supported previous studies, which showed that lifestyle-related behaviors tend to cluster in adolescence. In our study, most participants generally followed healthy lifestyle patterns, but we could also identify a group of young adults characterized by higher BMI (10% of all participants) in adolescence, as well as a group (16% of all participants) with more frequent co-occurrence of smoking, binge drinking, and low levels of physical activity in adolescence. We observed differences in biological aging between the classes characterized by adolescent lifestyle patterns in young adulthood, but the differences depended on the utilized epigenetic aging measure. Both the class with the overall unhealthiest lifestyle and that with a high BMI were biologically 1.7–3.3 years older than the classes with healthier lifestyle patterns when DNAm GrimAge was used to assess biological aging (AA$_{Grim}$). Moreover, they had 2–5 weeks/calendar year faster pace of biological aging (DunedinPoAm). The class with high BMI was biologically the oldest one when and DNAm PhenoAge and DunedinPACE were used. There were no differences when Horvath's and Hannum's clocks were used to estimate biological aging. The differences in lifestyle-related factors were maintained well over the transition from adolescence to young adulthood. However, genetic factors shared with adolescent lifestyle explained most of the observed differences in biological aging.

In our study, when the most recently published epigenetic aging measures were used, the class with the unhealthiest lifestyle was biologically 1.7–3.3 years older (AA$_{Grim}$) and had 2–3 weeks/calendar year faster pace of biological aging (DunedinPoAm) than the classes with healthier patterns. These measures can predict mortality and morbidity, especially cardiometabolic and lung diseases (*Belsky et al., 2020*; *Belsky et al., 2022*; *Lu et al., 2019*). A previous meta-analysis focusing on adults in a wide age range (17–99 years) showed that the number of healthy lifestyle behaviors is inversely associated with all-cause mortality risk (*Loef and Walach, 2012*). The mortality risk was up to 66% lower for individuals having multiple healthy behaviors compared to those adhering to an unhealthy lifestyle (smoking, low or high levels of alcohol use, unhealthy diet, no physical activity, and overweight). The accumulation of multiple unhealthy lifestyle habits during lifetime probably has a more detrimental effect on biological aging as well than any single lifestyle habit. However, our approach did not allow us to disentangle the effects of single lifestyle habits on biological aging. Our results suggest that the unhealthy lifestyle-induced changes in biological aging begin to accumulate in early life. These changes might predispose individuals to premature death in later life.

**Table 6.** The estimation results of the univariate model for biological aging among young adult twin pairs (MZ n = 154, DZ n = 211).

| | Model fit | | | | | | | | | Parameter estimates and their 95% confidence intervals | | | | | | | |
|---|---|---|---|---|---|---|---|---|---|---|---|---|---|---|---|---|---|
| | $X^2$ | df | SC | p | CFI | TLI | RMSEA | SRMR | BIC | $a^2$/total | | $c^2$ or $d^2$/total | | $e^2$/total | | Total | |
| **AA$_{Pheno}$** | | | | | | | | | | | | | | | | | |
| ACE | 5.2 | 3 | 1.27 | 0.155 | 0.98 | 0.99 | 0.06 | 0.06 | 2009 | 0.65 | 0.56, 0.74 | 0.00 | | 0.35 | 0.26, 0.45 | 1.00 | 0.89, 1.12 |
| ADE | 0.6 | 3 | 0.99 | 0.904 | 1.00 | 1.02 | 0.00 | 0.02 | 2003 | 0.03 | −0.46, 0.51 | 0.65 | 0.15, 1.15 | 0.33 | 0.25, 0.41 | 0.99 | 0.88, 1.09 |
| AE | 7.0 | 4 | 0.96 | 0.136 | 0.97 | 0.99 | 0.06 | 0.06 | 2003 | 0.65 | 0.56, 0.74 | - | | 0.35 | 0.26, 0.45 | 1.00 | 0.89, 1.12 |
| CE | 43.5 | 4 | 0.96 | <0.001 | 0.60 | 0.80 | 0.23 | 0.11 | 2038 | - | | 0.39 | 0.30, 0.48 | 0.61 | 0.52, 0.70 | 0.99 | 0.88, 1.10 |
| E | 107 | 5 | 0.96 | <0.001 | 0.00 | 0.59 | 0.33 | 0.21 | 2093 | - | | - | | 1.00 | | 0.99 | 0.88, 1.10 |
| **AA$_{Grim}$** | | | | | | | | | | | | | | | | | |
| ACE | 4.3 | 3 | 2.05 | 0.231 | 0.99 | 0.99 | 0.05 | 0.09 | 1989 | 0.73 | 0.66, 0.80 | 0.00 | | 0.27 | 0.20, 0.34 | 1.03 | 0.87, 1.20 |
| ADE | 5.6 | 3 | 1.55 | 0.133 | 0.98 | 0.98 | 0.07 | 0.09 | 1989 | 0.64 | 0.09, 1.19 | 0.09 | −0.48, 0.66 | 0.27 | 0.20, 0.34 | 1.03 | 0.87, 1.19 |
| AE | 5.7 | 4 | 1.54 | 0.220 | 0.98 | 0.99 | 0.05 | 0.09 | 1983 | 0.73 | 0.66, 0.80 | - | | 0.27 | 0.20, 0.34 | 1.03 | 0.87, 1.19 |
| CE | 33.0 | 4 | 0.87 | <0.001 | 0.72 | 0.86 | 0.20 | 0.12 | 2018 | - | | 0.50 | 0.40, 0.60 | 0.50 | 0.41, 0.60 | 1.02 | 0.87, 1.17 |
| E | 104 | 5 | 1.41 | <0.001 | 0.06 | 0.62 | 0.33 | 0.26 | 2115 | - | | - | | 1.00 | | 1.02 | 0.87, 1.17 |
| **DunedinPoAm** | | | | | | | | | | | | | | | | | |
| ACE | 1.3 | 3 | 1.12 | 0.722 | 1.00 | 1.02 | 0.00 | 0.04 | 2003 | 0.52 | 0.20, 0.85 | 0.09 | −0.20, 0.37 | 0.39 | 0.30, 0.48 | 0.98 | 0.86, 1.11 |
| ADE | 1.2 | 3 | 1.60 | 0.746 | 1.00 | 1.02 | 0.00 | 0.04 | 2003 | 0.62 | 0.53, 0.70 | 0.00 | | 0.38 | 0.30, 0.47 | 0.98 | 0.86, 1.10 |
| AE | 1.6 | 4 | 1.20 | 0.802 | 1.00 | 1.02 | 0.00 | 0.04 | 1997 | 0.62 | 0.53, 0.70 | - | | 0.38 | 0.30, 0.47 | 0.98 | 0.86, 1.10 |
| CE | 12.7 | 4 | 1.10 | 0.013 | 0.88 | 0.94 | 0.11 | 0.07 | 2009 | - | | 0.45 | 0.36, 0.55 | 0.55 | 0.45, 0.64 | 0.98 | 0.86, 1.10 |
| E | 85.1 | 5 | 1.15 | <0.001 | 0.00 | 0.55 | 0.30 | 0.22 | 2087 | - | | - | | 1.00 | | 0.98 | 0.86, 1.10 |
| **DunedinPACE** | | | | | | | | | | | | | | | | | |
| ACE | 2.0 | 3 | 1.08 | 0.582 | 1.00 | 1.00 | 0.00 | 0.05 | 1998 | 0.54 | 0.20, 0.87 | 0.08 | −0.21, 0.37 | 0.39 | 0.30, 0.48 | 0.99 | 0.87, 1.11 |
| ADE | 1.3 | 3 | 1.68 | 0.740 | 0.99 | 1.00 | 0.00 | 0.05 | 1981 | 0.42 | −0.13, 0.97 | 0.27 | −0.31, 0.84 | 0.32 | 0.24, 0.39 | 0.98 | 0.84, 1.13 |

*Table 6 continued on next page*

Table 6 continued

| | Model fit | | | | | | | | | Parameter estimates and their 95% confidence intervals | | | | | | | |
|---|---|---|---|---|---|---|---|---|---|---|---|---|---|---|---|---|---|
| | $X^2$ | df | SC | p | CFI | TLI | RMSEA | SRMR | BIC | a²/total | | c² or d²/total | | e²/total | | Total | |
| AE | 2.1 | 4 | 1.58 | 0.724 | 1.00 | 1.10 | 0.00 | 0.05 | 1976 | 0.68 | 0.52, 0.82 | - | | 0.32 | 0.24, 0.40 | 0.99 | 0.84, 1.15 |
| CE | 23.7 | 4 | 1.45 | <0.001 | 0.76 | 0.88 | 0.16 | 0.10 | 2007 | - | | 0.45 | 0.35, 0.54 | 0.55 | 0.46, 0.65 | 0.98 | 0.84, 1.13 |
| E | 78.5 | 5 | 1.47 | <0.001 | 0.09 | 0.64 | 0.28 | 0.23 | 2118 | - | | - | | 1.00 | | 0.98 | 0.84, 1.13 |

The epigenetic aging measures were adjusted for sex, age, and baseline pubertal development prior to analysis.

SC, scaling correction; CFI, comparative fit index; RMSEA, root mean square error of approximation; SRMR, standardized root-mean-square residual; BIC, Bayesian information criterion; MZ, monozygotic; DZ, dizygotic.

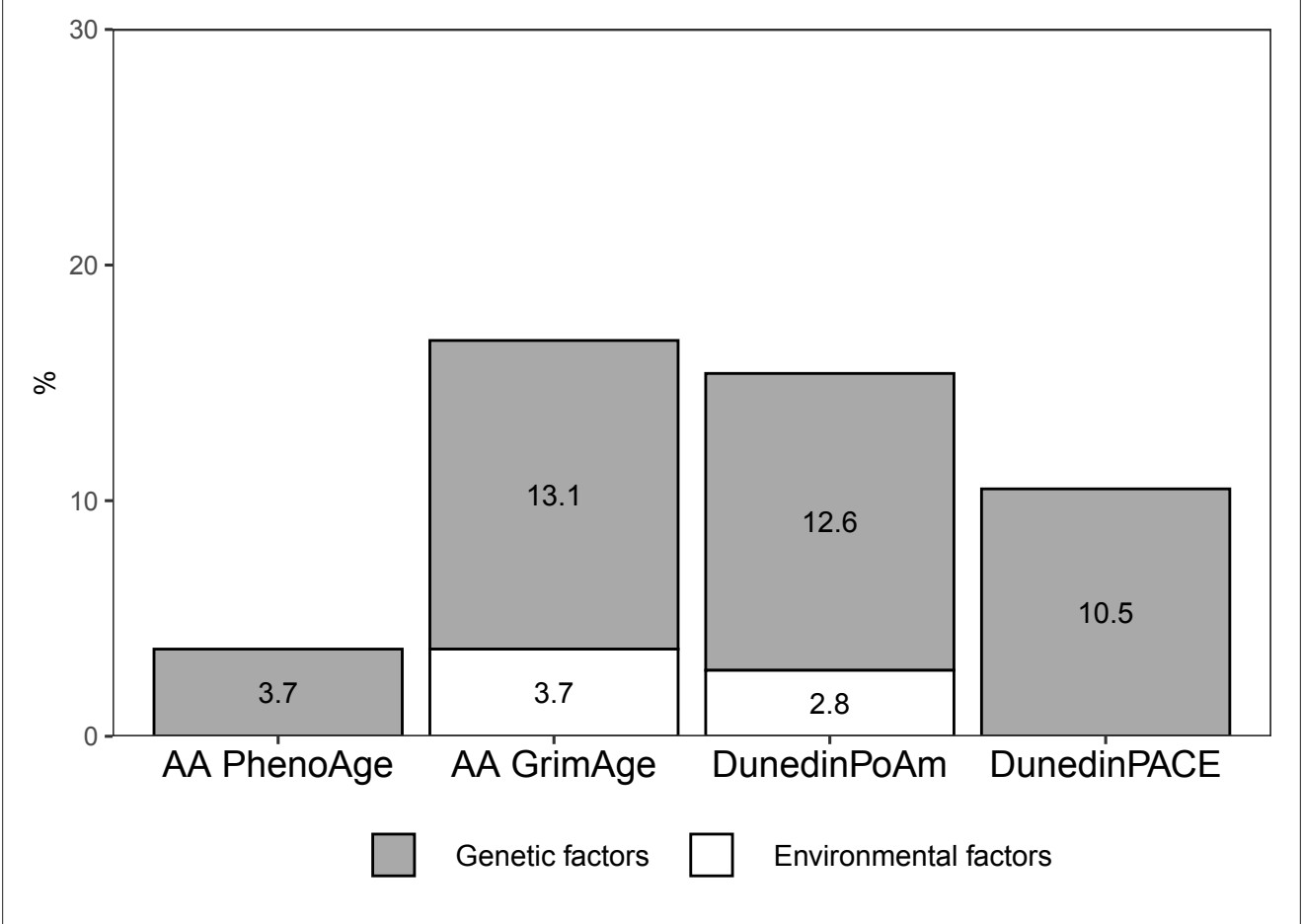

**Figure 4.** Proportions of the total variation in biological aging explained by genetic and (unshared) environmental factors shared with adolescent lifestyle patterns among young adult twin pairs (MZ n = 154, DZ n = 211). The results are based on the model including additive genetic and non-shared environmental component (AE model). AA, age acceleration.

The online version of this article includes the following source data for figure 4:

**Source data 1.** Genetic and environmental factors underlying the association between adolescent lifestyle patterns and biological aging.

To the best of our knowledge, this is the first study to investigate common genetic influences underlying lifestyle clusters and biological aging. Our results suggest genetic correlation between adolescent lifestyle and biological aging; individuals who are genetically prone to unhealthy lifestyles or overweight in adolescence are also susceptible to faster biological aging later in young adulthood. The shared genetic influences on two phenotypes may be due to several scenarios (*Solovieff et al., 2013*). They may arise from genetic pleiotropy; in this case, the genes may be a common cause for both adolescent lifestyle and biological aging. Another possible reason is causal relation between the phenotypes. In this case, genetic factors may affect adolescent lifestyle, which lies on the causal path to biological aging (or vice versa). However, for the relationship to be causal, it is necessary that there are shared environmental influences on the phenotypes (*De Moor et al., 2008*). In our study, environmental influences shared with adolescent lifestyle on biological aging were observed only when DNAm GrimAge and DunedinPoAm estimators were used. In line with our study, *McCartney et al., 2021* showed that there are shared underlying genetic contributions between single lifestyle factors and biological aging ($AA_{Grim}$, $AA_{Pheno}$) using polygenetic risk scores for epigenetic AA. Their Mendelian randomization analysis also suggested causal influences of BMI and smoking on biological aging, but only when DNAm GrimAge was used.

To the best of our knowledge, this is also the first study reporting the association between adolescent BMI (relative weight) and biological aging in later life. Previous systematic reviews have concluded that being overweight or obese in childhood and adolescence has a consistent impact on mortality

and morbidity in later life (*Park et al., 2012*; *Reilly and Kelly, 2011*). In particular, the associations with cardiometabolic morbidity are well-established, but the results of the studies investigating the associations independent of adult BMI are inconclusive (*Park et al., 2012*). A more recent study showed that early-life body size indirectly predisposes coronary artery disease and type 2 diabetes through body size in adulthood rather than having a direct effect (*Richardson et al., 2020*). Our results considering biological aging are in line with the existing literature but depend on the epigenetic clock utilized. In our study, the participants assigned to the class that was, on average, overweight in adolescence were biologically older (based on AA$_{Pheno}$, AA$_{Grim}$, DunedinPoAm, and DunedinPACE) in young adulthood compared to the classes of normal weight and healthy lifestyle habits. The group stood out, especially when AA$_{Pheno}$ and DunedinPACE were used to measure biological aging, but adult BMI explained the observed differences in these measures. Practically all variance of AA$_{Pheno}$ and DunedinPACE shared with adolescent lifestyle was explained by shared genetic factors. Therefore, these measures probably capture aspects of biological aging that are attributed to genetic factors shared with BMI. Mainly, the differences in AA$_{Grim}$ and DunedinPoAm did not attenuate after additionally controlling for adult BMI, suggesting that higher BMI in adolescence has a direct long-term effect on biological aging measured with these epigenetic clocks.

LTPA is associated with a lower risk of mortality and cardiovascular diseases (*Li et al., 2013*; *Löllgen et al., 2009*). Twin studies and genetically informed studies have suggested that genetic pleiotropy can partly explain these frequently observed associations (*Karvinen et al., 2015*; *Sillanpää et al., 2022*). Previous studies have shown that LTPA is also associated with slower biological aging (*Kankaanpää et al., 2021*). In this study, lower levels of physical activity in adolescence were closely intertwined with other unhealthy behaviors. To fully understand the role of adolescence physical activity in later biological aging would require a more comprehensive analysis of activity patterns, intensities, and modes, as well as subgroup analyses that account for other lifestyle factors, such as diet.

Adolescent smoking behavior and alcohol use appeared to be strongly clustered, in line with the findings of a recent systematic review (*Whitaker et al., 2021*). For this reason, the associations of smoking and alcohol use with biological aging might be difficult to disentangle. Smoking is the most detrimental lifestyle factor, and its association with accelerated biological aging has been frequently reported (*Oblak et al., 2021*). However, the results obtained for the association between alcohol use and biological aging remain unclear (*Oblak et al., 2021*). A recent study showed that smoking has a causal effect on AA$_{Grim}$, whereas alcohol use did not exhibit such effect (*McCartney et al., 2021*). Epigenetic methylation changes due to alcohol seem to be much fewer in number and magnitude compared to smoking exposure (*Stephenson et al., 2021*). In our study, the unhealthiest lifestyle class, in which smoking and alcohol use co-occurred, exhibited accelerated biological aging, especially when GrimAge and DunedinPoAm were used. These epigenetic aging measures are highly sensitive to tobacco exposure (*Belsky et al., 2020*; *Lu et al., 2019*). DNAm GrimAge is a composite biomarker comprising seven DNAm surrogates for plasma markers and smoking pack-years, which can predict the time to death (*Lu et al., 2019*). DunedinPoAm utilizes a specific CpG site (located within the gene AHRR), the methylation of which is strongly affected by tobacco exposure (*Belsky et al., 2020*). For these reasons, most of the variation in biological aging, which is explained by environmental factors shared with adolescent lifestyle, is probably due to smoking exposure.

To better understand the observed differences in biological aging, we also studied differences in DNAm-based surrogates included in the DNAm GrimAge estimator (*Figure 3—figure supplement 1*). Surprisingly, the class with the unhealthiest lifestyle pattern differed unfavorably from those with healthier habits only in DNAm-based smoking pack-years. The class with a high BMI had increased levels of several DNAm-based plasma markers, including DNAm PAI-1 and TIMP-1, which are associated with markers of inflammation and metabolic conditions (*Lu et al., 2019*). These findings support the suggestions that AA$_{Grim}$ is a useful biomarker for cardiovascular health and a potential predictor of cardiovascular disease already in young adulthood (*Joyce et al., 2021*).

Recent studies have yielded inconsistent results regarding the association between pubertal timing and biological aging (*Hamlat et al., 2021*; *Maddock et al., 2021*). In our models studying the differences in biological aging across adolescent lifestyle patterns, pubertal development at the age of 12 was not associated with accelerated biological aging in young adulthood (except for AA$_{Hannum}$). Moreover, the class with a high BMI included participants with advanced pubertal development, which might reflect the common genetic background underlying BMI and age at menarche (*Kaprio et al.,*

*1995*). All these findings support the studies showing that childhood obesity, which tracks forward into adulthood, explains the observed associations between advanced pubertal status and worse cardiovascular health (*Bell et al., 2018*) and can further reflect the genetic architecture underlying BMI, pubertal development, and worse health (*Day et al., 2015*).

Our study has the following major strengths. Adolescent lifestyle-related patterns were identified using population-based large cohort data (N ~ 5000), with longitudinal measurements of lifestyle-related factors assessed using validated questionnaires. Response rates were high and the distribution of the lifestyle-related patterns in the subsample of twins with information on biological aging was similar to the distribution in large cohort data, supporting the generalizability of our findings. Moreover, adolescent lifestyle behavior patterns were identified using data-driven LCA. This approach enabled us to use all available data on adolescent lifestyle-related behaviors and identify the patterns without using artificial cutoff points for the variables. The reciprocal associations between different lifestyle-related factors, as well as their joint association with biological aging, are complex, and individual associations are difficult to interpret. However, our approach produced results with easy interpretation. The data were prospective, and biological aging was assessed with novel epigenetic aging measures, including a recently published DunedinPACE estimator. Furthermore, for the first time, we could evaluate the proportions of genetic and environmental influences underlying adolescent lifestyle as a whole in relation to biological aging by using quantitative genetic modeling. However, our study also has some limitations. Adolescent lifestyle-related behaviors were self-reported and, therefore, might be susceptible to recall bias and bias through social desirability.

In conclusion, later biological aging reflects adolescent lifestyle behavior. Our findings advance research on biological aging by showing that a shared genetic background can underlie both adolescent lifestyle and biological aging measured with epigenetic clocks.

## Acknowledgements

This work was supported by the Academy of Finland (213506, 265240, 263278, 312073 to JK, 297908 to MO, and 341750, 346509 to ES), EC FP5 GenomEUtwin (JK), National Institutes of Health/National Heart, Lung, and Blood Institute (grant HL104125), EC MC ITN Project EPITRAIN (JK and MO), the University of Helsinki Research Funds (MO), Sigrid Juselius Foundation (JK and MO), Yrjö Jahnsson Foundation (6868), Juho Vainio Foundation (ES), and Päivikki and Sakari Sohlberg foundation (ES).

## Additional information

### Funding

| Funder | Grant reference number | Author |
|---|---|---|
| Academy of Finland | 213506 | Jaakko Kaprio |
| Academy of Finland | 297908 | Miina Ollikainen |
| Academy of Finland | 341750 | Elina Sillanpää |
| EC FP5 GenomEUtwin | | Jaakko Kaprio |
| National Institutes of Health | HL104125 | Jaakko Kaprio |
| EC MC ITN Project EPITRAIN | | Jaakko Kaprio |
| University of Helsinki Research Funds | | Miina Ollikainen |
| Sigrid Juselius Foundation | | Jaakko Kaprio |
| Yrjö Jahnsson Foundation | 6868 | Elina Sillanpää |
| Juho Vainio Foundation | | Elina Sillanpää |
| Päivikki and Sakari Sohlberg foundation | | Elina Sillanpää |

| Funder | Grant reference number | Author |
| --- | --- | --- |
| Academy of Finland | 312073 | Jaakko Kaprio |
| Academy of Finland | 263278 | Jaakko Kaprio |
| Academy of Finland | 265240 | Jaakko Kaprio |
| Academy of Finland | 346509 | Elina Sillanpää |

The funders had no role in study design, data collection and interpretation, or the decision to submit the work for publication.

## Author contributions

Anna Kankaanpää, Conceptualization, Formal analysis, Methodology, Writing – original draft; Asko Tolvanen, Supervision, Methodology, Writing – review and editing; Aino Heikkinen, Data curation, Writing – review and editing, Preprocessing of the DNAm data; Jaakko Kaprio, Elina Sillanpää, Conceptualization, Resources, Supervision, Funding acquisition, Project administration, Writing – review and editing; Miina Ollikainen, Conceptualization, Resources, Supervision, Funding acquisition, Writing – review and editing

## Author ORCIDs

Anna Kankaanpää http://orcid.org/0000-0002-6973-5385
Asko Tolvanen http://orcid.org/0000-0001-6430-8897
Aino Heikkinen http://orcid.org/0000-0001-5770-6475
Jaakko Kaprio http://orcid.org/0000-0002-3716-2455
Miina Ollikainen http://orcid.org/0000-0003-3661-7400
Elina Sillanpää http://orcid.org/0000-0001-6375-959X

## Ethics

Human subjects: Data collection was conducted in accordance with the Declaration of Helsinki. The Indiana University IRB and the ethics committees of the University of Helsinki and Helsinki University Central Hospital approved the study protocol (113/E3/2001and 346/E0/05). The blood samples for DNA analyses were collected during in-person clinical studies after written informed consent was signed.

## Decision letter and Author response

Decision letter https://doi.org/10.7554/eLife.80729.sa1
Author response https://doi.org/10.7554/eLife.80729.sa2

# Additional files

## Supplementary files

• Supplementary file 1. The codes used to analyse the data.
• MDAR checklist

## Data availability

A subsample of the FTC with DNA methylation age estimates, phenotypes, and information on the adolescent lifestyle behaviour patterns (BCH weights) will be located in the Biobank of the National Institute for Health and Welfare. All these data will be publicly available for use by qualified researchers following a standardised application procedure (see the website https://thl.fi/en/web/thl-biobank/for-researchers for details on the application process). Because of the consent given by study participants and the high degree of identifiability of the twin siblings in Finland, the full cohort data cannot be made publicly available. The full cohort data are available through the Institute for Molecular Medicine Finland (FIMM) Data Access Committee (DAC) for authorized researchers who have IRB/ethics approval and an institutionally approved study plan. For more details, please contact the FIMM DAC (fimm-dac@helsinki.fi). The codes used to analyse the data are provided in the supplementary file 1 and the processed data used to generate figures have been uploaded as the source data files.

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
