## [Editor Report]

This is an important article that is methodologically compelling that provides evidence that an unhealthy lifestyle during adolescence accelerates epigenetic age in adulthood and that these associations are largely explained by the effect of shared genetic influences. The main strengths of this article are the relatively large sample size, longitudinal assessment of lifestyle factors, and sophisticated statistical analyses. The article will be of interest to a broad audience, including individuals working on methylation, epidemiology, and/or aging.

---

## [Decision Letter]

**Decision letter after peer review:**

Thank you for submitting your article "The role of adolescent lifestyle habits in biological aging: A prospective twin study" for consideration by *eLife*. Your article has been reviewed by 3 peer reviewers, and the evaluation has been overseen by a Reviewing Editor and a Senior Editor. The following individuals involved in the review of your submission have agreed to reveal their identity: Jenny van Dongen (Reviewer #1); Esther Walton (Reviewer #3).

As is customary in *eLife*, the reviewers have discussed their critiques with one another. What follows below is the Reviewing Editor's edited compilation of the essential and ancillary points provided by reviewers in their critiques and in their interaction post-review. Please submit a revised version that addresses these concerns directly. Although we expect that you will address these comments in your response letter, we also need to see the corresponding revision clearly marked in the text of the manuscript. Some of the reviewers' comments may seem to be simple queries or challenges that do not prompt revisions to the text. Please keep in mind, however, that readers may have the same perspective as the reviewers. Therefore, it is essential that you attempt to amend or expand the text to clarify the narrative accordingly.

Essential revisions:

The reviewers provided a number of comments, mostly for providing clarity, which can be found below. One additional point that came up during the discussion is the use of the newly generated PCA-based epigenetic clocks created by Morgan Levine's group (https://www.nature.com/articles/s43587-022-00248-2), because this has been shown to improve the performance of some of the clocks used by the authors. However, given that it will likely take quite some additional work to recalculate the clocks based on this method, we decided to leave it up to you to reanalyze the data using the PCA-based clocks or just address this point in your Discussion section.

*Reviewer #1 (Recommendations for the authors):*

Introduction

Would it be possible to provide references for the following sentences?

Typically, about half of the adolescents fall into subgroups characterised by healthy lifestyle habits

However, small minorities of adolescents are classified as heavy substance users or as having multiple other risk behaviors

Methods

• It is mentioned that illumina 450k and illumina EPIC arrays were used. It might be informative to add the number of samples? Do I understand correctly that the raw data from the 2 platforms were QC'd and normalized together? That could be more explicitly mentioned.

• It is mentioned that the online age calculator was used. Could the authors please clarify what exactly was uploaded to the calculator? Do I understand correctly that pre-QC'd methylation β-values were used? Isn't the preferred method to upload raw methylation data because the online calculator does its own internal QC and normalization?

• In the introduction, it is mentioned that DunedinPACE is essentially an improvement of DunedinPoAm, but in the methods it is described that both measures are tested. Could the authors please clarify why they tested both measures? If these 2 clocks are supposed to measure the same thing, but DunedinPoAm is just outdated, why it still considered?

• The question on alcohol use at age 14 and 17 measures binge drinking, which is a different measure from the alcohol use measure at age 21-25. It is not clear why this measure was used. Was it the only measure of alcohol use available at the age of 14 and 17? Could the authors comment on how suited this measure is for quantifying alcohol use in this age group?

• On page 10, it is mentioned "the highest response category of the original questionnaire (development completed) was omitted for all items, except for menarche…". Could the authors please clarify what exactly was done in such cases, i.e. were these participants discarded from the analysis, or did they receive an NA (but was a sum score then still computed?), or were they recoded to the second-highest response category?

• How did the analysis that tested for mean differences in biological ageing between the lifestyle behavior patterns take account for the clustering of observations within families?

• The procedure to derive the proportion of variation in biological aging explained by genetic factors shared with adolescent lifestyle patterns was not entirely clear to me. For example, it is described that "On the other hand, it comprised the variance explained by the adolescent lifestyle patterns …". Is this sentence still referring to the same model as in the previous sentence? Perhaps it would help to clarify how many models were fitted.

• Could the authors clarify the rationale of the analysis that additionally corrected for adult BMI. I think the understandability of the paper could be improved if this would be explicitly explained in the methods and/or Results section.

• (How) did the authors correct for multiple testing?

Results/Discussion/interpretation

• My impression upon reading the Results section is that it seems to me (but maybe I'm wrong) that BMI is the main discriminative factor driving the 5 lifestyle classes. Could the authors comment on how they think about his, i.e. how much do the other variables; exercise, smoking and alcohol contribute? Is it possible to say something about this? Although I recognize that a major strength of the current analysis is that it make use of multiple lifestyle measures that are collectively summarized in one overall measure of lifestyle, I am left wondering how much each of the individual lifestyle measures is contributing and if we are mainly looking at the effect of BMI here. How different do the authors expect that the results would have been if the analysis would simply have been performed on BMI only?

• It is mentioned that group 4 shows differences in several DNAm-based plasma proteins, but group 5 only a difference in smoking pack-years, which seems a bit counter-intuitive, but no interpretation is given. Is the interpretation that group 5 appears to have the most unhealthy lifestyle but is biologically not the unhealthiest group? Or could this be a power issue (although group 5 I believe was larger than group 4)?

• It is concluded that (e.g. discussion p 19 and in the abstract): "Our findings suggest pleiotropic genetics effects; that is, the same genes affect both adolescent lifestyle and the pace of biological aging". However, pleiotropic effects are only one possible interpretation for shared genetic influences on two traits. Causal effects of lifestyle on biological aging, or vice versa of biological aging on lifestyle can also be captured in the shared genetic component (i.e. lifestyle in particular BMI is highly heritable, and if BMI has a causal effect on biological aging, this would give rise to a genetic correlation between these traits, i.e. rg includes both causal effects in both directions, as well as pleiotropy). The current analysis did not model or test causal paths. So I feel that the explanation of shared genetic influences should be more nuanced here.

• The discussion, on page 20 cites "previous meta-analysis showed that the number of healthy lifestyle behaviors is associated with all-cause mortality risk". Could information be added on the age of the populations in this study (is it also based on adolescents, as the current study is?)

• The discussion, on page 20, mentions "Our results suggested that the accumulation of multiple unhealthy lifestyle habits in adolescence has a more detrimental effect on biological aging that any single lifestyle habit." I have a hard time deducing on which results this conclusion is based. Did they also examine the effects of the single lifestyle habits? Or is this referring to previous work?

*Reviewer #3 (Recommendations for the authors):*

1) Can the authors also add (to table 2 or elsewhere) the correlations between chronological age and each epigenetic clock?

2) It wasn't fully clear why the authors adjust for adult BMI. What's the model / DAG here? Is adult BMI assumed to be a mediator or a downstream effect of epigenetic age? Line 481 suggests the authors consider mediation. However, I would assume that adolescent and adult BMI probably lie on a continuum and are largely correlated. So, why control for this association? I.e. what does it mean if adolescent BMI has a direct effect on adult ageing that is independent of adult BMI? What are the clinical implications? Another sentence on this or a DAG would help here.

3) 62-73% of explained variance through genetic factors seems large and 0-4% for (unshared) environmental factors seems very low. How does this compare to heritability reports in other studies? Is it likely that all variation is explained by genetics? Maybe this just needs some slight rephrasing also acknowledging the large amount of unexplained variation.

4) The authors write 'However, genetic factors shared with adolescent lifestyle explained most of the observed differences in biological aging'. I wasn't sure about the 'shared with adolescent lifestyle' phrase? Shouldn't this phrase be removed – or alternatively, please explain how exactly the link between genetics and lifestyle factors was assessed.

Related to this point, I wasn’t sure about the interpretation around pleiotropic genetic effects? The authors argue that they identified pleiotropic effects underlying BMI and epigenetic ageing. However, other scenarios are also possible, correct? E.g., even though any suggestions around causality should be done with extreme caution, it would be possible that genetic predictors for one trait (BMI, ageing) are on the causal pathway for the other trait. A brief discussion around this point might be helpful.

5) Independent lifestyle effects: there are two somewhat conflicting statements. In line 447, the authors argue that their ‘results suggested that the accumulation of multiple unhealthy lifestyle habits in adolescence has a more detrimental effect on biological aging than any single lifestyle habit’. However, in line 487+, they state that ‘lower levels of physical activity in adolescence were closely intertwined with other unhealthy behaviors. To fully understand the role of adolescence physical activity in later biological aging would require a more comprehensive analysis of activity patterns, intensities and modes, as well as subgroup analyses that account for other lifestyle factors, such as diet’. Maybe, these two separate statements could be more aligned.

6) Did the authors observe the same classes in the subgroup with DNAm data? Maybe Figure 2-suppl could be more in line with Figure 2 (same style). Or alternatively, combine a dot- and boxplot in Figure 2-suppl (and Figure 3). Also, please add to Figure 2-suppl the age at which each lifestyle factor was measured.

---

## [Author Response]

Essential revisions:The reviewers provided a number of comments, mostly for providing clarity, which can be found below. One additional point that came up during the discussion is the use of the newly generated PCA-based epigenetic clocks created by Morgan Levine’s group (https://www.nature.com/articles/s43587-022-00248-2), because this has been shown to improve the performance of some of the clocks used by the authors. However, given that it will likely take quite some additional work to recalculate the clocks based on this method, we decided to leave it up to you to reanalyze the data using the PCA-based clocks or just address this point in your Discussion section.

Thank you for this interesting suggestion. We have now calculated the epigenetic age estimates using principal component (PC)-based epigenetic clocks (PCHorvath, PCHannum, PCPhenoAge and PCGrimAge) created by Levine’s group (Higgins-Chen et al., 2022). The means of the epigenetic age estimates derived from the PC-based clocks were higher compared with those obtained with the original clocks, and the standard deviations were at the same level or even higher (Author response table 1). Also, several outliers (> 5 standard deviations away from mean) were detected according to PCA-based clocks (according to PCHannum 3 outliers, PCPhenoAge 4 and PCGrimAge 1) whereas no outliers were observed when original clocks were used. Recoding these observations to missing values did not result in considerably smaller standard deviations (Author response table 1). Moreover, the correlations between PCA-based age estimates and chronological age were even weaker than obtained with the original clocks (based on Horvath 0.13 vs. 0.17, Hannum 0.06 vs. 0.10, PhenoAge -0.07 vs. 0.03, and GrimAge 0.06 vs. 0.14). However, the correlations between epigenetic age acceleration (AA) measures assessed with different clocks were consistently higher when PC-based epigenetic clocks were used (Author response image 1).

**Author response table 1. sa2table1:** Means (standard deviations) of the epigenetic age estimates obtained using original and PC-based clocks.

	Original clock	PC clock	PC clock, outliers excluded
Horvath's clock	28.9 (3.6)	30.8 (3.8)	-
Hannum's clock	18.2 (3.3)	34.4 (4.0)	34.4 (3.8)
DNAm PhenoAge	13.0 (5.3)	16.8 (6.4)	16.6 (5.8)
DNAm GrimAge	25.2 (3.3)	38.8 (3.2)	38.8 (3.2)

**Author response image 1. sa2fig1:** Correlations between epigenetic age acceleration (AA) measures assessed with different clocks.

We also tested the differences between the adolescent lifestyle behavior patterns in biological aging assessed with PC-based epigenetic clocks and the results were very similar to the original ones. There were no differences in biological aging when PCHorvath (Wald test: *p* = 0.550) and PCHannum (*p* = 0.104) were used to assess biological aging, but differences were observed when PCPhenoAge (*p* = 0.031) PCGrimAge (*p* = 8.3e-8) were used (Author response image 2).

**Author response image 2. sa2fig2:** Mean differences between the adolescent lifestyle behavior patterns in biological aging measured with original and PC-based (**A**) DNAm PhenoAge and (**B**) DNAm GrimAge. The analysis was adjusted for sex (female), standardized age and baseline pubertal development. Means and 95% confidence intervals are presented. C1 = the class with the healthiest lifestyle pattern, C2 = the class with low–normal BMI, C3 = the class with a healthy lifestyle and high–normal BMI, C4 = the class with high BMI, C5 = the class with the unhealthiest lifestyle pattern. AA, age acceleration.

We have now mentioned this in the Results section (lines 409-418).

“In our study, high standard deviations of epigenetic age estimates were observed. Therefore, variation in AA measures may largely be attributable to technical variation, which is not biologically meaningful. Recently developed principal component (PC)-based clocks are shown to improve the reliability and validity of epigenetic clocks (Higgins-Chen et al., 2022). We therefore replicated our main analyses using PC-based epigenetic clocks (data not shown). The standard deviations of epigenetic age estimates were similar or even higher compared with those obtained with the original clocks, but the correlations between AA measures assessed with different clocks were consistently higher when PC-based epigenetic clocks were used. Importantly, the observed associations with the adolescent lifestyle behavior patterns did not substantially change.”

References:

Higgins-Chen, A. T., Thrush, K. L., Wang, Y., Minteer, C. J., Kuo, P.-L., Wang, M., Niimi, P., Sturm, G., Lin, J., Moore, A. Z., Bandinelli, S., Vinkers, C. H., Vermetten, E., Rutten, B. P. F., Geuze, E., Okhuijsen-Pfeifer, C., van der Horst, M. Z., Schreiter, S., Gutwinski, S., … Levine, M. E. (2022). A computational solution for bolstering reliability of epigenetic clocks: implications for clinical trials and longitudinal tracking. *Nature Aging*, *2*(7), 644–661. https://doi.org/10.1038/s43587-022-00248-2

Reviewer #1 (Recommendations for the authors):IntroductionWould it be possible to provide references for the following sentences?Typically, about half of the adolescents fall into subgroups characterised by healthy lifestyle habitsHowever, small minorities of adolescents are classified as heavy substance users or as having multiple other risk behaviors

We refer to a systematic review conducted by Whitaker and colleagues in the three consecutive sentences. We have now clarified this (lines 89-93).

Methods• It is mentioned that illumina 450k and illumina EPIC arrays were used. It might be informative to add the number of samples? Do I understand correctly that the raw data from the 2 platforms were QC'd and normalized together? That could be more explicitly mentioned.

We have now added the number of samples accordingly (lines 158-159)

“Of the samples included in this study, 744 samples were assayed using 450k and 80 samples using EPIC arrays.”

The raw data from different platforms was combined and preprocessed together. We have now mentioned this (lines 159-161).

• It is mentioned that the online age calculator was used. Could the authors please clarify what exactly was uploaded to the calculator? Do I understand correctly that pre-QC'd methylation β-values were used? Isn't the preferred method to upload raw methylation data because the online calculator does its own internal QC and normalization?

We normalized and conducted quality control for the methylation data prior to uploading data to the calculator, but we used normalization method implemented in the calculator, as well. The different normalization methods have different purposes. Our single-sample Noob normalization (ssNoob) is the best performing normalization method when data from the EPIC and 450k-arrays is integrated (Fortin et al., 2017). It is not affected by whether samples are combined or not before normalization. We also used Β-mixture quantile (BMIQ) normalization for correcting probe design bias (Teschendorff et al., 2013). According to the calculator tutorial, the purpose of the normalization method implemented in the calculator is to make data comparable to the training data of the epigenetic clock.

The function implemented in the calculator also makes some quality checks (produces predicted sex and tissue) that help to identify suspicious samples. However, own quality control is needed, as well.

References:

Fortin, J. P., Triche, T. J., & Hansen, K. D. (2017). Preprocessing, normalization and integration of the Illumina HumanMethylationEPIC array with minfi. *Bioinformatics*, *33*(4), 558–560. https://doi.org/10.1093/bioinformatics/btw691

Teschendorff, A. E., Marabita, F., Lechner, M., Bartlett, T., Tegner, J., Gomez-Cabrero, D., & Beck, S. (2013). A β-mixture quantile normalization method for correcting probe design bias in Illumina Infinium 450 k DNA methylation data. *Bioinformatics*, *29*(2), 189–196. https://doi.org/10.1093/bioinformatics/bts680

Calculator tutorial: https://horvath.genetics.ucla.edu/html/dnamage/TUTORIALonlineCalculator.pdf

• In the introduction, it is mentioned that DunedinPACE is essentially an improvement of DunedinPoAm, but in the methods it is described that both measures are tested. Could the authors please clarify why they tested both measures? If these 2 clocks are supposed to measure the same thing, but DunedinPoAm is just outdated, why it still considered?

We agree that DunedinPoAm estimator is outdated. However, both DunedinPoAm and DunedinPACE are very recently published. There are some dozens of studies applying DunedinPoAm while few have reported results based on DunedinPACE. We believe that information on DunedinPoAm is useful for some readers.

• The question on alcohol use at age 14 and 17 measures binge drinking, which is a different measure from the alcohol use measure at age 21-25. It is not clear why this measure was used. Was it the only measure of alcohol use available at the age of 14 and 17? Could the authors comment on how suited this measure is for quantifying alcohol use in this age group?

We also had information on the frequency of alcohol use at the age of 14 and 17 assessed with question ‘How often do you drink alcohol at all?’. However, this question does not take into account quantity of the alcohol used. Both measures have been widely used in studies that utilize data from the FinnTwin12 cohort (Rose et al., 2019). In Finland in the late 1990s, when members of the FinnTwin12 cohort were adolescents, alcohol use was increasingly drunkenness-oriented among 14-18 year-olds (Lintonen et al., 2000). Therefore, we think that the question on binge drinking is suitable measure for quantifying alcohol use in this population.

References:

Rose, R. J., Salvatore, J. E., Aaltonen, S., Barr, P. B., Bogl, L. H., Byers, H. A., Heikkilä, K., Korhonen, T., Latvala, A., Palviainen, T., Ranjit, A., Whipp, A. M., Pulkkinen, L., Dick, D. M., & Kaprio, J. (2019). FinnTwin12 Cohort: An Updated Review. *Twin Research and Human Genetics*, *22*(5), 302–311. https://doi.org/10.1017/thg.2019.83

Lintonen, T., Rimpelä, M., Ahlström, S., Rimpelä, A., & Vikat, A. (2000). Trends in drinking habits among Finnish adolescents from 1977 to 1999. *Addiction*, *95*(8), 1255–1263. https://doi.org/10.1046/j.1360-0443.2000.958125512.x

• On page 10, it is mentioned "the highest response category of the original questionnaire (development completed) was omitted for all items, except for menarche…". Could the authors please clarify what exactly was done in such cases, i.e. were these participants discarded from the analysis, or did they receive an NA (but was a sum score then still computed?), or were they recoded to the second-highest response category?

In the original questionnaire which included Pubertal developmental scale (PDS) developed by Petersen et al., there are four response categories: 1 = growth/change has not begun, 2 = growth/change has barely started and 3 = growth/change is definitely underway, and 4 = development complete (Petersen et al., 1988). The highest response option was removed already when designing the questionnaire, because completing the development was assumed to be very rare by the age of 12. Therefore, there were no such problematic cases in the data as described by the reviewer.

To clarify the text, we have now removed the following sentence:

“The highest response category of the original questionnaire (development completed) was omitted for all items, except for menarche.”

Interested readers may find further information from (Wehkalampi et al., 2008).

References:

Petersen, A. C., Crockett, L., Richards, M., & Boxer, A. (1988). A self-report measure of pubertal status: Reliability, validity, and initial norms. *Journal of Youth and Adolescence*, *17*, 117–133.

Wehkalampi, K., Silventoinen, K., Kaprio, J., Dick, D. M., Rose, R. J., Pulkkinen, L., & Dunkel, L. (2008). Genetic and environmental influences on pubertal timing assessed by height growth. *American Journal of Human Biology*, *20*(4), 417–423. https://doi.org/10.1002/ajhb.20748

• How did the analysis that tested for mean differences in biological ageing between the lifestyle behavior patterns take account for the clustering of observations within families?

We used TYPE=MIXTURE COMPLEX option of the ANALYSIS command in conjunction with the CLUSTER=family option of the VARIABLE command in Mplus (Muthén & Muthén, 1998-2018). This approach corrects the standard errors for non-independence of observations within families. We have briefly mentioned this in the end of the statistical analysis section (lines 304-305).

• The procedure to derive the proportion of variation in biological aging explained by genetic factors shared with adolescent lifestyle patterns was not entirely clear to me. For example, it is described that "On the other hand, it comprised the variance explained by the adolescent lifestyle patterns …". Is this sentence still referring to the same model as in the previous sentence? Perhaps it would help to clarify how many models were fitted.

Thank you for raising this question. We noticed that the Figure 1 was unintentionally dropped out from the full submission. We have now included it in the manuscript, and hopefully, that makes the description of the analysis easier to follow. We have also clarified the paragraph which describes the estimation of genetic and environmental influences (lines 283-300).

• Could the authors clarify the rationale of the analysis that additionally corrected for adult BMI. I think the understandability of the paper could be improved if this would be explicitly explained in the methods and/or Results section.

Thank you for this excellent suggestion. We have now briefly clarified the rationale in the Results section (lines 397-408).

“According to the previous literature, it is controversial whether childhood obesity has a direct effect on later health, or whether the association is fully mediated by BMI in adulthood (Park et al., 2012). The role of adult BMI may depend on which disease outcome is studied (Richardson et al., 2020). After additionally adjusting the model for BMI in adulthood, the differences in AA_Pheno_ and DunedinPACE between the class of participants with high BMI (C4) and those with lower BMI (C1, C2, C5) were attenuated (Table 5, M2). This finding suggests that the observed differences in biological aging probably are fully mediated by BMI in adulthood. However, the differences in biological aging were only slightly attenuated when the DNAm GrimAge and DunedinPoAm estimators were used, suggesting that childhood overweight may leave permanent imprint on biological aging assessed with these measures.”

• (How) did the authors correct for multiple testing?

To control for Type I error rate due to multiple testing we have now calculated the 99% confidence intervals for the mean differences presented in the Table 2 and pointed out the differences that exist after controlling for multiple testing. Mainly, the interpretation of the results remained the same but the difference in biological aging between the unhealthiest and the healthiest pattern (C5 vs C1) was not significant at 0.01 level when DunedinPACE was used.

Results/Discussion/interpretation• My impression upon reading the Results section is that it seems to me (but maybe I'm wrong) that BMI is the main discriminative factor driving the 5 lifestyle classes. Could the authors comment on how they think about his, i.e. how much do the other variables; exercise, smoking and alcohol contribute? Is it possible to say something about this? Although I recognize that a major strength of the current analysis is that it make use of multiple lifestyle measures that are collectively summarized in one overall measure of lifestyle, I am left wondering how much each of the individual lifestyle measures is contributing and if we are mainly looking at the effect of BMI here. How different do the authors expect that the results would have been if the analysis would simply have been performed on BMI only?

We agree that BMI may have a dominant role in the extraction of the patterns. However, if the classification was based only on BMI, we would have missed the separate classes of participants with generally healthy lifestyle (C1) and participants with unhealthiest lifestyle pattern (C5). The participants in both classes were of normal weight on average, but there were remarkable differences in lifestyle habits between the classes. There were also clear differences between the classes in biological aging measured with AA_Grim_ and DunedinPoAm estimators.

Based on the reported analyses, we cannot say how much individual lifestyle habits, including leisure-time physical activity, smoking and alcohol use contribute to the classification. Probably physical activity and alcohol use have only a minor role, but interestingly, we found that low levels of physical activity and binge drinking co-occured with smoking.

• It is mentioned that group 4 shows differences in several DNAm-based plasma proteins, but group 5 only a difference in smoking pack-years, which seems a bit counter-intuitive, but no interpretation is given. Is the interpretation that group 5 appears to have the most unhealthy lifestyle but is biologically not the unhealthiest group? Or could this be a power issue (although group 5 I believe was larger than group 4)?

Surprisingly, we did not observe any increased levels of DNAm-based plasma proteins in young adulthood in the class with the unhealthiest lifestyle pattern compared to the classes with healthier habits (Figure 3—figure supplement 1). Therefore, we think that lack of differences is not due to power issue. Probably, the consequences of the unhealthy lifestyle will be seen also in the DNAm-based plasma proteins in later adulthood if unhealthy habits persist.

• It is concluded that (e.g. discussion p 19 and in the abstract): "Our findings suggest pleiotropic genetics effects; that is, the same genes affect both adolescent lifestyle and the pace of biological aging". However, pleiotropic effects are only one possible interpretation for shared genetic influences on two traits. Causal effects of lifestyle on biological aging, or vice versa of biological aging on lifestyle can also be captured in the shared genetic component (i.e. lifestyle in particular BMI is highly heritable, and if BMI has a causal effect on biological aging, this would give rise to a genetic correlation between these traits, i.e. rg includes both causal effects in both directions, as well as pleiotropy). The current analysis did not model or test causal paths. So I feel that the explanation of shared genetic influences should be more nuanced here.

Thank you for this important note. We agree that our interpretation of the shared genetic influences was too straightforward. We have now included more in-depth discussion on the possible reasons for the shared genetic influences (lines 490-496).

“The shared genetic influences on two phenotypes may be due to several scenarios (Solovieff et al., 2013). They may arise from genetic pleiotropy; in this case, the genes may be a common cause for both adolescent lifestyle and biological aging. Another possible reason is causal relation between the phenotypes. In this case, genetic factors may affect adolescent lifestyle, which lies on the causal path to biological aging (or vice versa). However, for the relationship to be causal, it is necessary that there are shared environmental influences on the phenotypes (de Moor et al., 2008).”

• The discussion, on page 20 cites "previous meta-analysis showed that the number of healthy lifestyle behaviors is associated with all-cause mortality risk". Could information be added on the age of the populations in this study (is it also based on adolescents, as the current study is?)

The age range of the populations in this meta-analysis was wide ranging from 17 to 99 years, but majority were older adult populations. Probably due to the lack of data, adolescents were not included in the study. We have now added the age range of the target populations in the manuscript (lines 475-476).

• The discussion, on page 20, mentions "Our results suggested that the accumulation of multiple unhealthy lifestyle habits in adolescence has a more detrimental effect on biological aging that any single lifestyle habit." I have a hard time deducing on which results this conclusion is based. Did they also examine the effects of the single lifestyle habits? Or is this referring to previous work?

We agree that based on our results, we cannot make this kind of conclusion. We did not conduct comparisons of the effects of the single lifestyle habits vs. the effect of overall lifestyle on biological aging. Therefore, we have now rephrased (lines 480-485):

“The accumulation of multiple unhealthy lifestyle habits during lifetime probably has a more detrimental effect on biological aging, as well, than any single lifestyle habit. However, our approach did not allow us to disentangle the effects of single lifestyle habits on biological aging. Our results suggest that the unhealthy lifestyle-induced changes in biological aging begin to accumulate in early life. These changes might predispose individuals to premature death in later life.”

Reviewer #3 (Recommendations for the authors):1) Can the authors also add (to table 2 or elsewhere) the correlations between chronological age and each epigenetic clock?

Due to the narrow age range of this study (21 to 25 years), the correlations were low. The correlations between epigenetic age and chronological age ranged from 0.03 to 0.17. Considerably higher estimates (ranging from 0.54 to 0.76) were observed in our previous study among participants in a wider age range (21 to 42 years) (Kankaanpää et al., 2021), which also included the participants of the current study. In the current study, chronological age did not correlate with pace of aging measured with DunedinPoAm and even slightly negative correlation (-0.09) was observed with DunedinPACE.

References:

Kankaanpää, A., Tolvanen, A., Saikkonen, P., Heikkinen, A., Laakkonen, E. K., Kaprio, J., Ollikainen, M., & Sillanpää, E. (2022). Do Epigenetic Clocks Provide Explanations for Sex Differences in Life Span? A Cross-Sectional Twin Study. *The Journals of Gerontology: Series A*, 77(9): 1898-1906. https://doi.org/10.1093/gerona/glab337

2) It wasn't fully clear why the authors adjust for adult BMI. What's the model / DAG here? Is adult BMI assumed to be a mediator or a downstream effect of epigenetic age? Line 481 suggests the authors consider mediation. However, I would assume that adolescent and adult BMI probably lie on a continuum and are largely correlated. So, why control for this association? I.e. what does it mean if adolescent BMI has a direct effect on adult ageing that is independent of adult BMI? What are the clinical implications? Another sentence on this or a DAG would help here.

Thank you for raising this question. We agree that this was unclear. We have now provided rationale for the adjustment in the result section (lines 397-408).

“According to the previous literature, it is controversial whether childhood obesity has a direct effect on later health, or whether the association is fully mediated by BMI in adulthood (Park et al., 2012). The role of adult BMI may depend on which disease outcome is studied (Richardson et al., 2020). After additionally adjusting the model for BMI in adulthood, the differences in AA_Pheno_ and DunedinPACE between the class of participants with high BMI (C4) and those with lower BMI (C1, C2, C5) were attenuated (Table 5, M2). This finding suggests that the observed differences in biological aging probably are fully mediated by BMI in adulthood. The differences in biological aging were only slightly attenuated when the DNAm GrimAge and DunedinPoAm estimators were used, suggesting that childhood overweight may leave permanent imprint on biological aging assessed with these measures.”

3) 62-73% of explained variance through genetic factors seems large and 0-4% for (unshared) environmental factors seems very low. How does this compare to heritability reports in other studies? Is it likely that all variation is explained by genetics? Maybe this just needs some slight rephrasing also acknowledging the large amount of unexplained variation.

In our study, genetic factors explained 62-73% of the total variation in biological aging depending on the utilized estimator, and unshared environmental factors accounted the rest of the variation (27-38%). Previous studies using twin and pedigree data have obtained moderate to high heritability estimates for age acceleration (AA) measures (Horvath & Raj, 2018). Our heritability estimates are somewhat higher than previously reported for AA measures based on DNAm PhenoAge (0.33-0.41) (Jylhävä et al., 2019; Levine et al., 2018) and DNAm GrimAge (0.30-0.58) (Lu et al., 2019; Lundgren et al., 2022). We are not aware of any studies reporting on the heritability of pace of aging measured with DunedinPoAm and DunedinPACE.

The reported proportions 0-4% reflect the size of environmental correlation between adolescent lifestyle and biological aging i.e. to which extent shared environmental factors explain both adolescent lifestyle and biological aging (see also answers above). Because of the nominal scale of the latent class variable, we were not able to directly estimate the genetic and environmental correlations using traditional bivariate twin modeling. Therefore, we evaluated these proportions indirectly. We have now clarified the description of the estimation of genetic and environmental influences (lines 283-300) and the presentation of the results (lines 437-449).

References:

Horvath, S., & Raj, K. (2018). DNA methylation-based biomarkers and the epigenetic clock theory of ageing. *Nature Reviews Genetics*, *19*(6), 371–384. https://doi.org/10.1038/s41576-018-0004-3

Levine, M. E., Lu, A. T., Quach, A., Chen, B. H., Assimes, T. L., Bandinelli, S., Hou, L., Baccarelli, A. A., Stewart, J. D., Li, Y., Whitsel, E. A., Wilson, J. G., Reiner, A. P., Aviv, A., Lohman, K., Liu, Y., Ferrucci, L., & Horvath, S. (2018). An epigenetic biomarker of aging for lifespan and healthspan. *Aging*, *10*(4), 573–591. https://doi.org/10.18632/aging.101414

Jylhävä, J., Hjelmborg, J., Soerensen, M., Munoz, E., Tan, Q., Kuja-Halkola, R., Mengel-From, J., Christensen, K., Christiansen, L., Hägg, S., Pedersen, N. L., & Reynolds, C. A. (2019). Longitudinal changes in the genetic and environmental influences on the epigenetic clocks across old age: Evidence from two twin cohorts. *EBioMedicine*, *40*, 710–716. https://doi.org/10.1016/j.ebiom.2019.01.040

Lu, A. T., Quach, A., Wilson, J. G., Reiner, A. P., Aviv, A., Raj, K., Hou, L., Baccarelli, A. A., Li, Y., Stewart, J. D., Whitsel, E. A., Assimes, T. L., Ferrucci, L., & Horvath, S. (2019). DNA methylation GrimAge strongly predicts lifespan and healthspan. Aging, 11(2), 303–327. https://doi.org/10.18632/aging.101684

Lundgren, S., Kuitunen, S., Pietiläinen, K. H., Hurme, M., Kähönen, M., Männistö, S., Perola, M., Lehtimäki, T., Raitakari, O., Kaprio, J., & Ollikainen, M. (2022). BMI is positively associated with accelerated epigenetic aging in twin pairs discordant for BMI. *Journal of Internal Medicine*. https://doi.org/10.36684/33-2020-1-680-686

4) The authors write 'However, genetic factors shared with adolescent lifestyle explained most of the observed differences in biological aging'. I wasn't sure about the 'shared with adolescent lifestyle' phrase? Shouldn't this phrase be removed – or alternatively, please explain how exactly the link between genetics and lifestyle factors was assessed.Related to this point, I wasn't sure about the interpretation around pleiotropic genetic effects? The authors argue that they identified pleiotropic effects underlying BMI and epigenetic ageing. However, other scenarios are also possible, correct? E.g., even though any suggestions around causality should be done with extreme caution, it would be possible that genetic predictors for one trait (BMI, ageing) are on the causal pathway for the other trait. A brief discussion around this point might be helpful.

Thank you for these important notes. We have now removed the phrase as suggested. We have also clarified the description of the estimation of genetic and environmental influences (lines 283-300). We also noticed that the Figure 1 was unintentionally dropped out from the full submission. We have now included it in the manuscript, and hopefully, that makes the description of the analysis easier to follow.

We must admit that our interpretation around pleiotropic genetic effects was too straightforward. We have now briefly discussed this topic (lines 490-496):

“The shared genetic influences on two phenotypes may be due to several scenarios (Solovieff et al., 2013). It may arise from genetic pleiotropy; in this case, the genes may be a common cause for both adolescent lifestyle and biological aging. Another possible reason is causal relation between the phenotypes. In this case, genetic factors may affect adolescent lifestyle, which lies on the causal path to biological aging (or vice versa). However, for the relationship to be causal, it is necessary that there are shared environmental influences on the phenotypes (de Moor et al., 2008).“

5) Independent lifestyle effects: there are two somewhat conflicting statements. In line 447, the authors argue that their 'results suggested that the accumulation of multiple unhealthy lifestyle habits in adolescence has a more detrimental effect on biological aging than any single lifestyle habit'. However, in line 487+, they state that 'lower levels of physical activity in adolescence were closely intertwined with other unhealthy behaviors. To fully understand the role of adolescence physical activity in later biological aging would require a more comprehensive analysis of activity patterns, intensities and modes, as well as subgroup analyses that account for other lifestyle factors, such as diet'. Maybe, these two separate statements could be more aligned.

We thank you for raising this point. We agree that these statements are somewhat conflicting. We did not conduct comparisons of the effects of the single lifestyle habits vs. the effect of overall lifestyle on biological aging. For that reason, we have modified the first statement (lines 480-485):

“The accumulation of multiple unhealthy lifestyle habits during lifetime may have a more detrimental effect on biological aging, as well, than any single lifestyle habit. However, our approach did not allow us to disentangle the effects of single lifestyle habits on biological aging. Our results suggest that the unhealthy lifestyle-induced changes in biological aging begin to accumulate in early life. These changes might predispose individuals to premature death in later life.”

6) Did the authors observe the same classes in the subgroup with DNAm data? Maybe Figure 2-suppl could be more in line with Figure 2 (same style). Or alternatively, combine a dot- and boxplot in Figure 2-suppl (and Figure 3). Also, please add to Figure 2-suppl the age at which each lifestyle factor was measured.

Initially, we conducted LCA in the subgroup of participants with DNAm data. The BIC value reached its minimum value at the 4-class-solution, and the interesting class of participants with high BMI (C4) was extracted at fifth step. Using large cohort data, we were able to confirm that the pattern C4 actually exists. The mean and probability profiles of the 5-class solution were very similar to those obtained with large cohort data. The class-sizes were C1: 34.5%, C2: 20.2%, C3: 19.2%, C4: 6.6%, and C5: 19.5% (the class-sizes in the subsample based on the LCA-solution using large cohort data: C1: 33.0%, C2: 16.6%, C3: 20.6%, C4: 10.1%, C5: 19.7%).

We have now modified the Figure 2 —figure supplement 1 to be more in line with Figure 2 and added the age at which each lifestyle factor was measured.